

# Subglacial and subaerial fluvial sediment transport capacity respond differently to water discharge variations

Ian Delaney[1], Andrew J. Tedstone[1,2], Mauro A. Werder[3,4], and Daniel Farinotti[3,4]

[1]Institut des dynamiques de la surface terrestre (IDYST), Université de Lausanne, Bâtiment Géopolis, 1015 Lausanne, Switzerland

[2]Department of Geosciences, University of Fribourg, Ch. du Musée 1700, Fribourg, Switzerland

[3]Laboratory of Hydraulics, Hydrology and Glaciology (VAW), ETH-Zürich, Hönggerbergring 26, 8093 Zürich, Switzerland

[4]Swiss Federal Institute for Forest, Snow and Landscape Research (WSL) Züricherstrasse 111, 8903 Birmensdorf, Switzerland

**Correspondence:** Ian Delaney (IanArburua.Delaney@unil.ch)

**Abstract.** Sediment transport capacity in both subaerial and subglacial channels depends on the shear stress exerted across the channel bottom, which varies with water velocity and channel width. In subaerial channels, water discharge variations are accommodated by flow depth and width changes, along with water velocity. However, in subglacial channels, water is pressurized by the ice above, and they grow in response to frictional heating of water flowing through them. As a result, water discharge changes mainly result in velocity variations, as the channel geometry evolves slowly (over days). Here, we present formulations of sediment transport capacity in different channel types and apply subglacial and subaerial hydraulics models to hydrographs from an Alpine glacier and the Greenland Ice. Numerical experiments show that the changing channel size results in sediment transport capacity peaking before the maximum water discharge. This hysteresis in channel size causes a highly variable relationship between sediment and water discharge in a transport-limited subglacial system. The results also indicate that high subglacial sediment transport capacities can occur across a wide range of water discharges. A second set of numerical experiments shows that subglacial sediment transport is highly non-linear with respect to water discharge, creating more variability in sediment transport capacity. Yet, results and formulations of subglacial sediment transport capacity show that its variability can approach that of subaerial systems when subglacial channel size is in equilibrium with water discharge. The implications of these findings are discussed in the context of sediment discharge from glaciers with different hydro-climatic forcings. We also discuss the impact of different assumptions of channel behavior on sediment transport capacity. These findings can improve the interpretation of sediment discharge records in glacierized catchments.

## 1 Introduction

Changes in glacier dynamics and hydrology have motivated numerous recent studies on sediment transport processes in cold regions (e.g. Li et al., 2022; Vergara et al., 2022; Zhang et al., 2022). Increases in sediment transport have been observed in Greenland (Bendixen et al., 2017), the European Alps (Costa et al., 2018), the Himalayas (Li et al., 2021), and the Andes (Vergara et al., 2022). To accurately explain observed changes in sediment transport in glacierized catchments, the processes



controlling sediment discharge and its variations with water discharge need to be examined (e.g. Riihimaki et al., 2005; Swift et al., 2005).

Glacier abrasion and quarrying sculpt landscapes, and create sediment that is transported fluvially over periods of millennia
or longer (c.f. Hallet, 1979; Iverson, 2012; Ugelvig et al., 2018). Pressurized subglacial water can transport this sediment from underneath glaciers (Walder and Fowler, 1994; Creyts et al., 2013; Beaud et al., 2018; Delaney et al., 2019) if it is reachable by the water.

In a transport-limited regime, sediment discharge is controlled by sediment transport capacity, which is defined as the amount of sediment the water can carry. In both subglacial and subaerial channels, sediment transport capacity depends on the shear
stress between water and the sediment it flows over (Shields, 1936; Meyer-Peter and Müller, 1948; Engelund and Hansen, 1967), along with the width of the channel bottom $w$ over which sediment mobilizes. The shear stress $\tau$ responds to the velocity of water $v$ flowing through the channel so that

$$\tau \propto v^2. \tag{1}$$

Following mass conservation, the mean velocity of the water flowing through a channel is

$$v = \frac{Q}{S}, \tag{2}$$

where $Q$ is water discharge, and $S$ is the channel's wetted area.

In subaerial channels operating with open channel flow, $S$ evolves with changing water discharge $Q$, by changing both the channel width and the water depth (Leopold and Maddock, 1953). The change in water depth results in a proportional increase in water velocity, and the shear stress $\tau$ increases according to Equation 1.

The response of water velocity to changing water discharge in subglacial channels differs from subaerial ones, however. The size of subglacial channels is controlled by the opposing processes of channel opening by frictional heating of water flow on
the one hand, versus creep closure by ice flow on the other (Röthlisberger, 1972). As a result, the subglacial channel size only evolves relatively slowly over days, whereas water discharge can vary more quickly over hours (e.g. Iken and Bindschadler, 1986; Andrews et al., 2014; Nanni et al., 2020). Therefore, subglacial water flow behaves more like pipe flow over short periods (hours, days). Changes in water discharge $Q$ are mainly accommodated by changing water velocity $v$ (Equation 2 and Figure 1; Alley et al., 1997).

Because of the above, sediment mobilization in subaerial and subglacial channels responds differently to changing water discharge. These differences are implicitly included in a range of models quantifying sediment transport in both subglacial and subaerial channels (e.g. Walder and Fowler, 1994; Alley et al., 1997; Tucker and Slingerland, 1997; Creyts et al., 2013; Beaud et al., 2018; Delaney et al., 2019; Hewitt and Creyts, 2019; Wickert and Schildgen, 2019). To date, several modeling frameworks examine subglacial sediment transport with evolving channel size (Creyts et al., 2013; Beaud et al., 2018; Delaney
et al., 2019; Hewitt and Creyts, 2019). However, these works minimally discuss the impact of different hydrological regimes on sediment transport capacity underneath glaciers in the context of interpreting sediment transport records. The few explicit parameterizations of subglacial sediment transport capacity with respect to water discharge assume fixed channel size (Alley





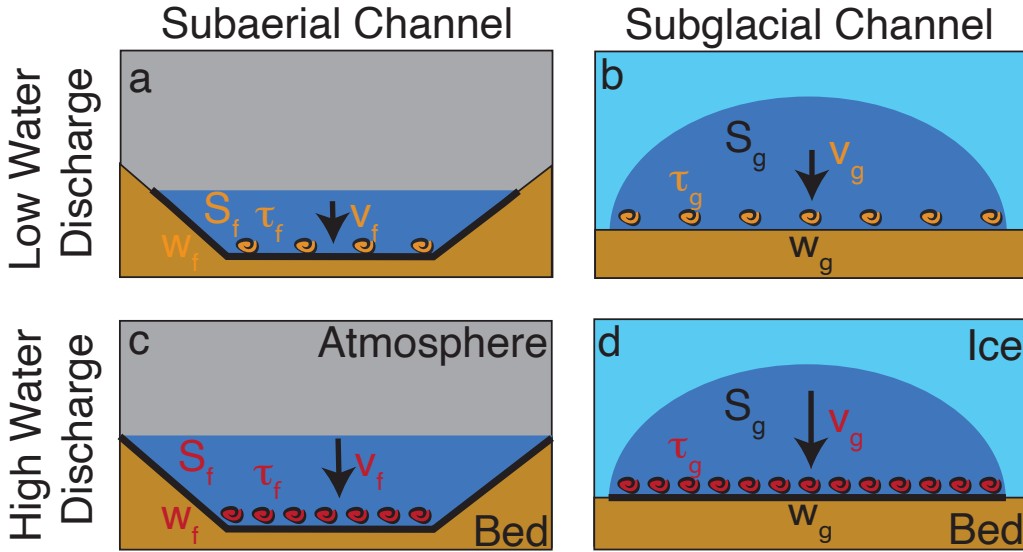

**Figure 1.** Sketch for the different responses of subglacial and subaerial channels to increased water discharge over short time scales. Arrow length denotes water velocity magnitudes in the subglacial (subaerial) $v_g$ ($v_f$) channels. $S_g$ ($S_f$) represents the wetted area in subglacial (subaerial) channels. The subglacial channel width $w_g$ remains unchanged, while the subaerial channel width $w_f$ evolves with water discharge. Subglacial (subaerial) shear stress $\tau_g$ ($\tau_f$) is responsible for the mobilization of sediment.

et al., 1997). These formulations demonstrate a strongly non-linear response in subglacial sediment transport capacity to water discharge. Yet, the continually evolving channels' size can impact variations in sediment transport capacity. Understanding

these processes is imperative for establishing the effect of hydro-climatic conditions on subglacial sediment dynamics, especially as sediment discharge capacity controls the mobilization and deposition of sediment. This makes it a fundamental aspect of subglacial sediment evacuation, especially over short timescales.

Despite the different physical processes between the subaerial and subglacial systems, observed water discharge and sediment export are often compared in contemporary glacierized catchments (e.g. Willis et al., 1996; Hodson et al., 1998; Pearce

et al., 2003; Richards and Moore, 2003; Swift et al., 2005; Chu et al., 2009; Tedstone and Arnold, 2012; Chu et al., 2012; Overeem et al., 2017; Delaney et al., 2018; Swift et al., 2021; Lu et al., 2022; Andresen et al., 2024). Many of these studies discuss the disparity between water discharge and sediment export, especially given the role of sediment access or production. Yet, a variable relationship between water discharge and sediment export could be expected from glaciers with evolving conduit size in a transport-limited regime, where sediment transport responds to the channel's hydraulic conditions, instead of

sediment availability. The response of sediment transport capacity to water discharge variations may affect the interpretation of sediment transport records in these glacierized catchments (e.g. Ganti et al., 2016; Mancini et al., 2023). This relationship must be clarified as sediment dynamics change in glacierized regions along with changing hydrology (e.g. Brunner et al., 2019),



increasing variability of water discharge (Lane and Nienow, 2019) and occurrence of extreme glacier melt (e.g. Overeem et al., 2015; Cremona et al., 2023) that may drive high magnitude sediment transport events.

This manuscript has two objectives: 1) to establish whether sub-seasonal water discharge can co-vary with sediment transport capacity in subglacial systems, and 2) to evaluate the variability of sediment transport capacity in subglacial channels with evolving channel size and different hydrographs compared to subaerial ones. We use numerical models to examine sediment transport capacity in both subglacial channels that evolve in size and subaerial channels that remain fixed. They are applied to proglacial hydrological records from an Alpine glacier in Switzerland (Fieschergletscher) and a land-terminating glacier in Greenland (Leverett Glacier). Model outputs demonstrate the specific processes that can drive variability in sediment discharge capacity in subglacial systems. We also run a model ensemble to evaluate the sediment transport capacity variability in subglacial channels compared to subaerial channels. Lastly, we present algebraic formulations of the sediment transport capacity response to water discharge in subaerial, steady-state R-channel, and pipe flow conditions. These formulations extend the relationships presented in Alley et al. (1997) and illustrate the sediment transport capacity behavior of different channel assumptions. Findings indicate differences in the relationships amongst water discharge, channel geometry, water velocity and sediment transport capacity in subglacial and subaerial channels. The manuscript then discusses the implications of variability in sediment transport capacity from glaciers and the interpretation of sediment transport records.

## 2 Study sites and data

Water discharge data is from Alpine and ice sheet settings collected downstream of the glacier, and assumes no water storage in the proglacial area. The Alpine site (labeled *ALPINE*) is Fieschergletscher in the Swiss Alps ($46°\,29'\,07''$ N, $8°\,08'\,3''$ E). The water discharge data used here was collected at a $1\,\mathrm{min}$ interval from May 24, 2014, to October 10, 2014 (Figure 2 a Felix et al., 2022).

The Leverett Glacier in Greenland (labeled *ICESHEET*) serves as the ice sheet setting. Water discharge was measured roughly $2\,\mathrm{km}$ downstream from the terminus ($67°\,03'\,5''$ N, $50°\,12'\,59''$ W), at a $5\,\mathrm{min}$ time interval from May 28, 2012 to August 8, 2012 Tedstone et al. (2013, Figure 2 b).

## 3 Methods

The two models described below (Sections 3.1 and 3.2) represent relationships amongst water discharge, water velocity, and channel geometry in both subaerial and subglacial channels (Table 1). Both models use the measured discharge to calculate water velocity, shear stress, and width-integrated shear stress, upon which both suspended sediment and bedload transport depend (Figure 1; Shields, 1936). Our choice to evaluate our results in terms of shear stress omits the selection of a sediment transport relationship and a grain-size parameter (e.g. Shields, 1936; Meyer-Peter and Müller, 1948).





### 3.1 Subglacial channel model

The subglacial channel model accounts for the channel geometry and the water's velocity to evaluate the shear stress of water flowing across sediments underneath a glacier. We use a lumped hydraulics model from Werder et al. (2010), itself based upon Clarke (1996).


Here, it is assumed that the water is transported through a subglacial channel (Figure 1; Röthlisberger, 1972) beneath a glacier with channel length $l$, with a flat bed and a mean thickness of $h_{ice}$. The channel size grows from melt due to frictional heating from water flow and closes due to ice creep. The formulation here does not consider the englacial storage of water. The evolution of subglacial channel size $S_g$ is given as

$$\frac{\partial S_g}{\partial t} = C_1 \frac{Q \Delta h}{l} - C_2 \left( h_o - \frac{\Delta h}{2} \right)^n S_g, \tag{3}$$

where $t$ is time, $C_1 = (1 - \rho_w c_p c_t) \frac{\rho_w g}{\rho_i L}$ and $C_2 = 2A(\frac{\rho_w g}{n})^n$ are constants (values in Table 1), $g$ is the acceleration due to gravity, $Q$ is water discharge, $\Delta h$ is the hydraulic head drop change over $l$, $h_o = \frac{\rho_i}{\rho_w} h_{ice}$ is the mean ice overburden pressure expressed in meter water equivalent ($\rho_w$ is density of water; $\rho_i$ is density of ice), and $n$ is Glen's n (usually $n = 3$; Glen, 1955). The first term on the equation's right side represents the channel opening by frictional heating, while the following term represents channel closure from ice deformation.

Following the Darcy-Weisbach equation, the head drop $\Delta h$ is

$$\Delta h = l \frac{1}{2g} f_i \frac{v_g^2}{D_h}, \tag{4}$$

where $f_r$ is a friction factor, $D_h$ is the hydraulic diameter, $l$ is the channel length, and $v_g = \frac{Q}{S_g}$ is the water velocity. The hydraulic diameter $D_h$ is converted to wetted area $S_g$ with

$$S_g = \frac{D_h^2}{2} \frac{(\frac{\beta}{2} + \sin \frac{\beta}{2})^2}{\beta - \sin \beta}, \tag{5}$$

where $\beta$ is the central angle of the circular segment that comprises the channel (the Hooke angle, Hooke et al. (1990)). $\beta = \pi$

corresponds to a semi-circular channel and smaller values of $\beta$ result in shallow, wide channels. This completes the subglacial hydraulic model which is described by the state variables $S_g$ and $\Delta h$.

The shear stress, $\tau_g$, between the water and the channel bed results from the Darcy-Weisbach formulation

$$\tau_g = \frac{1}{8} f_r \rho_w v_g^2, \tag{6}$$

where $v_g = \frac{Q}{S_g}$ is the water velocity. The width of the channel floor $w_g$ is represented as

$$w_g = 2 \sin \frac{\beta}{2} \sqrt{\frac{2 S_g}{\beta - \sin \beta}}. \tag{7}$$

This value establishes the integrated shear stress across the channel $w_g \tau_g$.



## 3.2 Subaerial channel model

The hydraulics parameterization presented in Tucker and Slingerland (1997) is implemented to represent the subaerial channel. This model uses mass conservation and the Darcy-Weisbach relationship and assumes that the channel is sufficiently wide compared to its depth. This latter assumption means that the hydraulic radius is well approximated by the flow depth. Therefore, the resulting shear stress $\tau_f$ at the river bed is

$$\tau_f = \frac{\rho_w \, g^{\frac{2}{3}} \, f_f^{\frac{1}{3}}}{2} \left(\frac{Q}{w_f}\right)^{\frac{2}{3}} \nabla z_c^{\frac{2}{3}}, \tag{8}$$

where $\nabla z_c$ is the channel slope, and $f_f$ is the friction factor for subaerial channels (Tucker and Slingerland, 1997). Channel width $w_f$ is

$$w_f = k Q^\alpha, \tag{9}$$

where $k$ is a constant and $\alpha = \frac{1}{3}$ is a commonly chosen exponent (Leopold and Maddock, 1953). Also following the Darcy-Weisbach, subaerial water velocity, $v_f$, is given as

$$v_f = \sqrt{\frac{8 \tau_f}{f_f \, \rho_w}}. \tag{10}$$

As mentioned above, the width-integrated shear stress is $w_f \tau_f$.

Note that this subaerial channel model is purely algebraic, whereas the subglacial model comprises a differential equation for the evolution of $S_g$. Thus, the channel size in the subaerial model has no history dependence on the discharge $Q_w$, whereas the subglacial one does (Equation 3).



**Table 1.** Variables, parameters, and constants used in this work. Where two values are given, the first refers to *ALPINE* , a scenario from Fieschergletscher, and the second to a glacier marginal to the *ICESHEET*  a scenario from Leverett Glacier. A second line refers to the range of values examined in the parameter search.

| Name | Symbol | Value (*ALPINE*  or *ICESHEET* ) | Units |
|---|---|:---:|---|
| **Variables** | | | |
| Water discharge | $Q$ | | $\mathrm{m^3\,s^{-1}}$ |
| Water velocity (subglacial, subaerial) | $v, (v_g, v_f)$ | | $\mathrm{m\,s^{-1}}$ |
| Channel wetted area (subglacial, subaerial) | $S_g, S_f$ | | $\mathrm{m^2}$ |
| Channel depth (subaerial) | $H$ | | m |
| Hydraulic diameter | $D_h$ | | m |
| Width of channel floor (subglacial, subaerial) | $w, (w_g, w_f)$ | | m |
| Hydraulic head | $\Delta h$ | | m |
| Hydraulic gradient | $\Psi = \frac{\Delta h}{l}$ | | $\mathrm{m\,m^{-1}}$ |
| Shear stress (subglacial, subaerial) | $\tau, (\tau_g, \tau_f)$ | | $\mathrm{Pa\,m^{-2}}$ |
| Stream power | $\Omega$ | | $\mathrm{kg\,m\,s^{-3}}$ |
| **Parameters and Constants** | | | |
| Gravitational constant | $g$ | 9.81 | $\mathrm{m\,s^{-2}}$ |
| Density of water | $\rho_w$ | 1000 | $\mathrm{kg\,m^{-3}}$ |
| Density of ice | $\rho_i$ | 900 | $\mathrm{kg\,m^{-3}}$ |
| Hooke angle of channel | $\beta$ | $\frac{\pi}{6}$ $(\frac{\pi}{10}, \pi)$ | rad |
| Friction factor (subglacial, subaerial) | $f, (f_r, f_f)$ | -(5, (16, 3)) (0.01, 21) | $(-)$ |
| Glacier thickness | $h_{ice}$ | 225 or 740 | m |
| Effective glacier thickness | $h_o$ | $\frac{\rho_i}{\rho_w} h_{ice}$ | m |
| Effective glacier length | $l$ | $7{,}000$ or $26{,}000$ | m |
| Constant 1 in Equation 3 | $C_1$ | $2.2 \times 10^{-5}$ | $\mathrm{m^{-1}}$ |
| Constant 2 in Equation 3 | $C_2$ | $3.7 \times 10^{-13}$ | $\mathrm{m^{-n}\,s^{-1}}$ |
| Latent heat of fusion | $L$ | 333.5 | $\mathrm{kJ\,kg^{-1}}$ |
| Pressure melting coefficient | $c_t$ | $7.5 \times 10^{-8}$ | $\mathrm{K\,Pa^{-1}}$ |
| Specific heat capacity of water | $c_p$ | 4180 | $\mathrm{J\,kg^{-1}K^{-1}}$ |
| Ice flow constant | $A$ | $5.3 \times 10^{-24}$ | $\mathrm{Pa^{-n}\,s^{-1}}$ |
| Ice flow exponent | $n$ | 3 | $(-)$ |
| Gradient of channel bed (subaerial) | $\nabla z_c$ | 0.02 (.01, 0.05) | $(-)$ |
| Subaerial channel factor | $k$ | 8 | $\mathrm{s\,m^{-2}}$ |
| Channel geometry exponent | $\alpha$ | $\frac{1}{3}$ $(\frac{1}{3}, \frac{1}{2})$ | $(-)$ |



### 3.3 Implementation

The models above are applied to proglacial discharge records from the Fieschergletscher (scenario *ALPINE*) and the Leverett Glacier (scenario *ICESHEET*). The model outputs represent generalizable sediment transport characteristics from these hy-

drographs, rather than actual hydraulic conditions. To generalize these scenarios, *ALPINE* is exemplified by relatively thin ice thickness ($h_{ice}$= 225 m Grab et al., 2021), low water discharge ($\sim 10\,\mathrm{m^3\,s^{-1}}$) and high diurnal variability in water discharge at Fieschergletscher (Figure 2 a). *ICESHEET* is exemplified by thick ice ($h_{ice}$= 700 m; Morlighem et al., 2017), high water discharge ($\sim 300\,\mathrm{m^3\,s^{-1}}$) and low diurnal variability in water discharge at Leverett glacier (Figure 2 e).

In the first experiment, the models are applied to a reference test case for each glacier. These experiments assumes a sub-

glacial channel with $\beta = \frac{\pi}{6}$ and a subaerial channel with $\alpha = \frac{1}{3}$ and slope of $0.02$ (Table 1). Both friction factors $f_r$ and $f_f$ are tuned so that reasonable water velocities ($\sim 1 - 1.5\,\mathrm{m\,s^{-1}}$ Werder et al., 2010; Chandler et al., 2013) occur for both *ALPINE* and *ICESHEET*. To test the covariance between sediment transport capacity and water discharge, model outputs are compared to water discharge from the glaciers using Spearman rank correlation. This metric accounts for the ordering of values, but not their magnitude. Rank correlation reduces the impact of the non-linear relationship between sediment transport capacity and

hydrology.

The second experiment aims to characterize the variability in sediment discharge capacity in subglacial relative to subaerial channels across a range of channel slopes and shapes, and friction factors. Additionally, we examine the effects of differing water discharge smoothing periods from $15\,\mathrm{min}$ up to $15\,\mathrm{days}$. The results below present water velocity ($v_g$, $v_f$), shear stress ($\tau_g$, $\tau_f$), and width-integrated shear stress ($w_g\tau_g$, $w_f\tau_f$) from the subglacial and subaerial models. For brevity, these values

together are referred to as "model outputs" in the text. To evaluate variability, we subtract the model outputs from their daily averages. This creates a time series that is detrended from the seasonal variations in the model outputs and has an approximately normal distribution. We then present the variability in the model output by taking the standard deviation of this detrended time series.

The effects of channel shape, gradient, and roughness are established by running the model with random parameter values

of channel slope and geometry factors ($\beta$, $\nabla z_c$, and $\alpha$) and friction factors ($f_r$ and $f_f$; see Table 1 for value range). Runs with parameter combinations are accepted if their mean subglacial water velocity over the season lies between $0.5\,\mathrm{m\,s^{-1}}$ and $2\,\mathrm{m\,s^{-1}}$ or if subaerial water velocity lies between $0.3\,\mathrm{m\,s^{-1}}$ and $1.2\,\mathrm{m\,s^{-1}}$ (e.g. Werder et al., 2010; Magnusson et al., 2012; Chandler et al., 2013). To be accepted, subglacial model runs must experience a flotation fraction ($\frac{\Delta h}{2h_o}$) of above $1.2$ for less than $2.5\,\%$ of the run. A model spinup dictates the initial condition of cross-sectional area $S_g$. The spinup consists of applying

the maximum observed water discharge of the first $4$ days of the study period to the model until there is no change in the channel area $S_g$.

The routine runs until $100$ different parameter combinations for each water discharge smoothing period are accepted with the conditions described above. To test the effects of variability in water discharge, it is averaged over the smoothing period ranging




## 4 Results

### 4.1 Changing subglacial channel size drives different timing and variability in sediment transport capacity

The first numerical experiment aims to quantify the sources of increased variability in the subglacial model outputs as they exhibit different seasonal evolutions and peaks (Figure 2). Variable relationships between model outputs and water discharge emerge for the subaerial and subglacial cases due to the hysteresis in channel size in the subglacial model (Equation 3). Because subaerial channels have no history dependence, each water discharge value in the subaerial channel produces a unique water velocity, shear stress, and width-integrated shear stress (Section 3.1). This characteristic results in a perfect rank correlation between the variables and water discharge (Figure 3). An inconsistent relationship between subglacial model outputs and water discharge persists across the range of parameters examined in the ensemble runs (see Section 4.2; Figure S15).

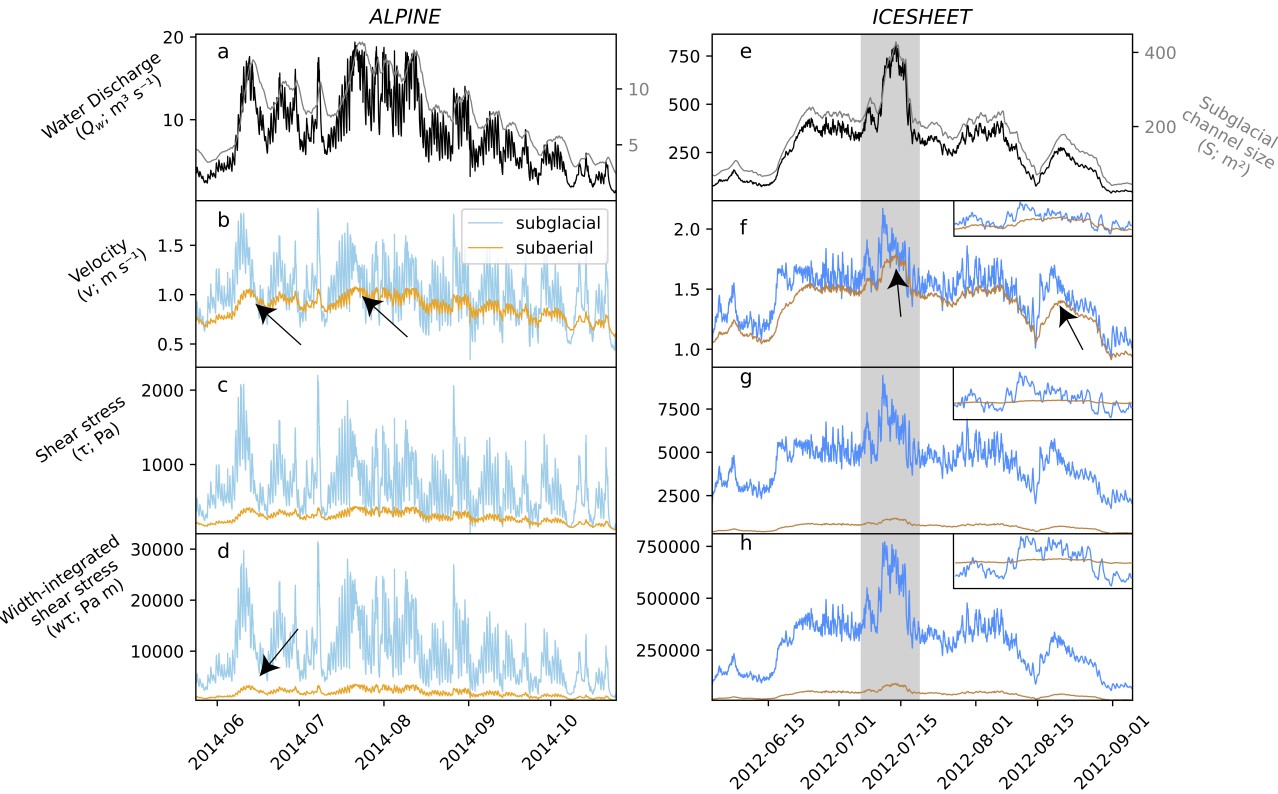

**Figure 2.** Model outputs from simulations using the hydrographs in panels a and e for the scenarios *ALPINE* (a-d) and *ICESHEET* (e-h). Black (gray) lines in a and e represent the subglacial channel size (water discharge). Blue (orange) lines represent outputs from the subglacial (subaerial) channel. Data are shown at 15min intervals. Arrows show examples where variables peak in the subglacial channel before the subaerial one. Insets in f-h show the peak melt event denoted by the shaded area in panels e–h, with an arbitrary y-axis.





Peaks in subaerial model outputs occur coincident with peaks in water discharge (Figure 2). In the subglacial channel,
peaks in model outputs generally occur when water discharge increases at the fastest rate, but before the maximum water
discharge. As the water discharge stabilizes at its peak, channel growth continues, causing water velocity and other model
outputs to decrease from their peak values (Section 3.1). As a result, subglacial sediment transport capacity is greatest on the
hydrograph's rising limb, relative to the falling limb, creating a hysteresis effect.

The history dependence on channel size in subglacial channels means that different sediment transport characteristics, such
as velocity, occur across a large range of water discharges. For instance, in subglacial channels in *ALPINE* , high water velocity
values and shear stresses can occur from a low water discharge ($\sim 4\,\mathrm{m^3\,s^{-1}}$) to the maximum water discharge at over 17
$\mathrm{m^3\,s^{-1}}$ (Figure 3 a). In *ICESHEET* , water velocities close to the seasonal mean value can occur at water discharges between
roughly 150 $\mathrm{m^3\,s^{-1}}$ and 310 $\mathrm{m^3\,s^{-1}}$. The subglacial channel's evolving width can counteract some of these effects. Width-
integrated shear stress generally increases with water discharge, with greater rank correlation compared to water velocity
or shear stress (*ALPINE* , Figure 3 a–c). Yet, even the width-integrated shear stress can vary substantially relative to water
discharge. The highest values of width-integrated shear stress occur at water discharge values ranging from roughly 11 $\mathrm{m^3\,s^{-1}}$
to over 17 $\mathrm{m^3\,s^{-1}}$. The variability in width-integrated shear stress is less pronounced in the *ICESHEET* scenario, where the
hydrograph has less diurnal variability (Figure 3 c, f). This results from the discharge variations being in closer equilibrium
with subglacial channels compared to *ALPINE* (Sections 4.3 and 4.2).

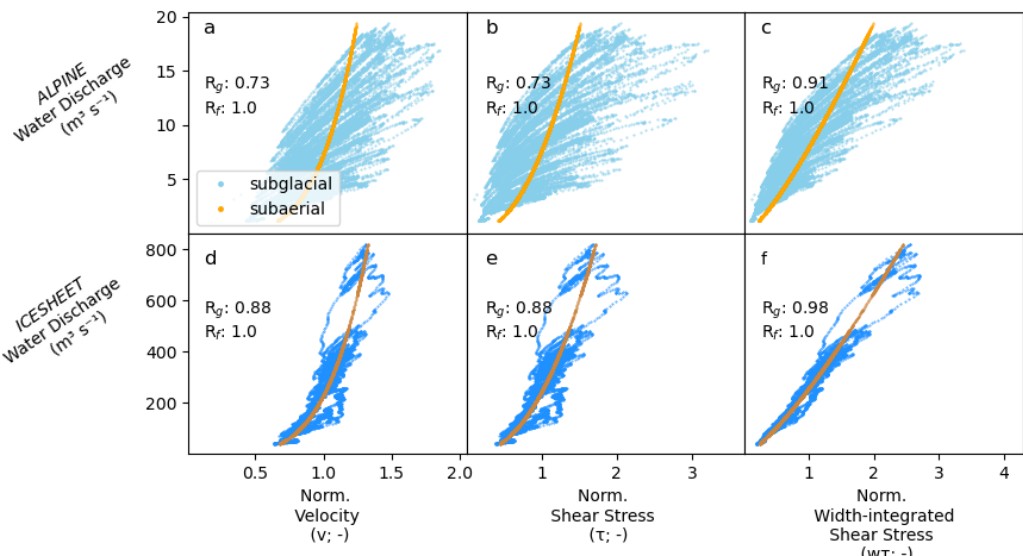

**Figure 3.** Relationship between water discharge and normalized velocity, shear stress, and width-integrated shear stress for *ALPINE* (a-c)
and *ICESHEET* (d-f). Variables on the x-axis have been normalized to mean values. $R_g$ ($R_f$) shows the Spearman rank correlations for the
subglacial (subaerial) outputs. Plots are shown with discharge and model outputs at 15min intervals.



## 4.2 Increased variability in evolving subglacial channels occurs across a range of channel shapes, slopes, and friction values

The second numerical experiment aims to compare the variability between the subglacial and subaerial model outputs to a range of channel shapes, friction factors, and water discharge variability. This is accomplished by applying a range of parameter values to models applied to the different hydrological regimes and examining the variability in model outputs (Section 3.3).

Across the range of parameters examined, variability in all model outputs (i.e. velocity, shear stress, and width-integrated shear stress) remains higher in the subglacial system compared to the subaerial one for both *ALPINE* and *ICESHEET* (Figure 4). In some cases, subglacial model outputs' variability is a magnitude larger than their subaerial counterparts. Variability in both subglacial and subaerial outputs decreases with smoothing time longer than approximately 1–5 days (Figure 4). These smoothing timescales remove the diurnal variations in water discharge, thereby reducing variability in model outputs. Velocity variability in *ICESHEET* is substantially smaller than *ALPINE* (Figure 4 a and d). This could result from the subglacial conduit being in closer equilibrium with water discharge. Variations in *ICESHEET* shear stress are comparable to *ALPINE* due to the effects of the evolution of shear stress with subglacial conduit size (Equations 3 and 12;). The much greater water flux and thus subglacial channel size in *ICESHEET* drives the larger width integrated shear stress variations (Figure 2 c and f).

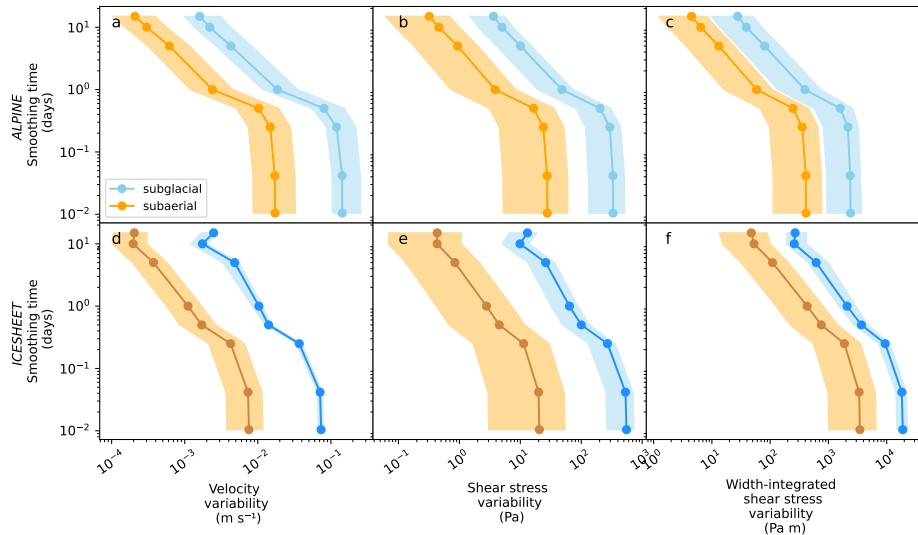

**Figure 4.** Standard deviation of detrended model outputs for different smoothing times with different subglacial and subaerial channel shapes and friction factors. Shaded areas denote the range of standard deviations from the accepted 100 parameter combinations (Section 3.3). Solid lines denote the mean value of standard deviations. Markers show smoothing periods (15 min, 1 hr, 6 hr 12 hr, 1 d, 5 d, 10 d, and 15 d).

Greater subglacial friction factors $f_i$ result in greater variability in the shear stress and width-integrate shear stress in the *ALPINE* case (Figure 5). This result is expected given Equations 6. Low values of $f_i$ result in slower growth rates in subglacial channels (Equations 3 and 4). As a result, water velocity, as opposed to channel growth, accommodates increases in water





discharge, and velocity variability increases. Higher values of $f_i$ allow faster channel growth that accommodates increases in water discharge, reducing velocity variability. Smaller values of channel factor $\beta$, creating low and broad channels, result in more variability in width-integrated shear stress. Here, the channel width can grow more quickly in response to water discharge

increases as compared to a semi-circular channel with $\beta = \pi$ (Equation 4).

Smaller values of subaerial channel shape factor $\alpha$, or a channel cross-section shape closer to a slot canyon, result in greater variability in subaerial velocity and shear stress that can approach the variability of subglacial values (Figure 5). Steeper subaerial channel slopes $\nabla z_c$ result in greater variability in both shear stress and width-integrated shear stress but only approach the subglacial channel's variability in the *ALPINE* case (Figure 5). Greater values of subaerial friction factors $f_p$ result in greater

variability of shear stress and width-integrated shear stress. Yet, they do not exceed the variability of subglaical channels. We note that these parameter values span a commonly accepted range. Therefore, we do not anticipate a scenario where variability in the subaerial system would exceed the subglacial system with these two hydrologic forcings.

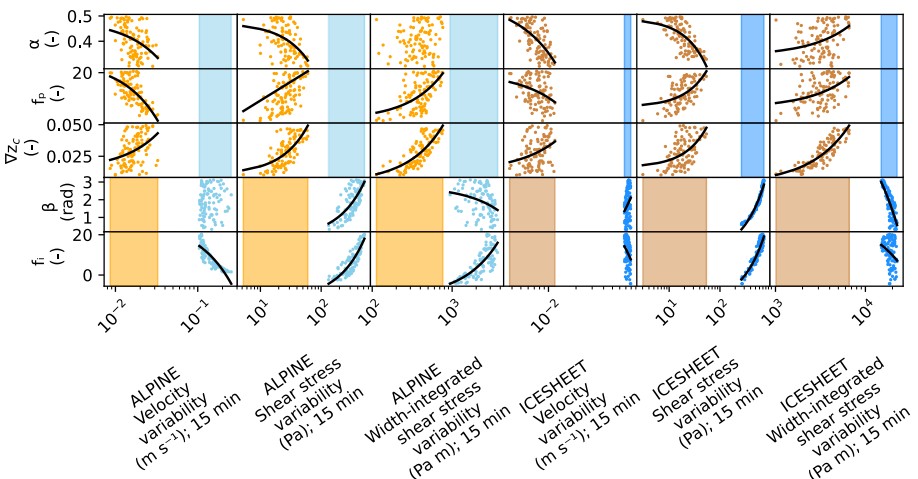

**Figure 5.** Parameter values compared to variability in model outputs. Plots on the right (orange, light blue) correspond to the *ALPINE* case. Plots on the left (brown, dark blue) correspond to the *ICESHEET* case. Outputs are shown using 15 minute smoothing time. Linear trends are denoted in black if they are significant at $p \geq 0.01$. To compare the range of model outputs for the subaerial and subglacial cases, shaded areas show range of model outputs that are independent of the parameter values.

## 4.3   Sediment transport scaling in different channel types

The numerical experiments above consider the size evolution of subglacial channels and demonstrate that for these hydrographs

subglacial sediment transport variability is greater than its subaerial counterpart (Section 4.2). Here, we compare the sediment transport behavior of different channel types as they respond to water discharge, channel shape, and hydraulic gradient. This





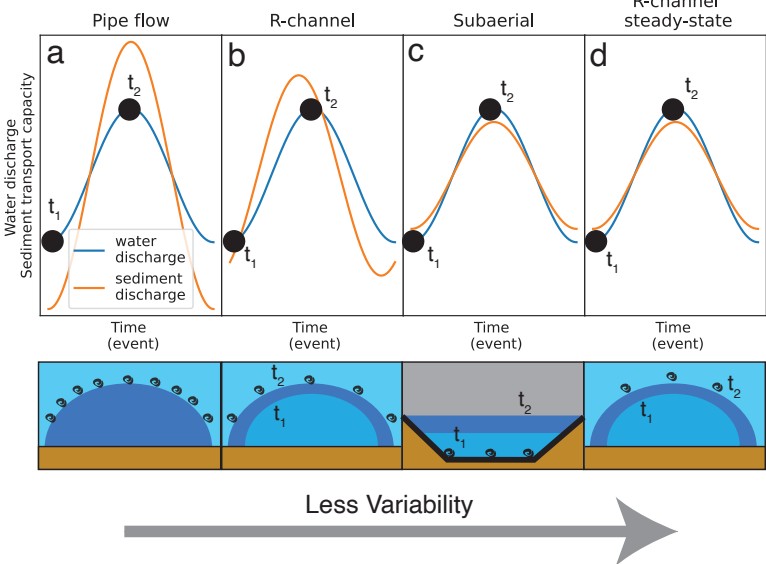

**Figure 6.** Response of sediment transport capacity (orange line) and channel size (bottom row) to a water discharge event (blue line) under different channel conditions. a) Pipe flow with fixed channel size described in Section 4.3. b) R-channel described in Section 3.1. c) Subaerial channel described in Section 3.2. d) Steady-state R-channel (channel size evolves in equilibrium with water discharge) described in Section 4.3. Note axes are not to scale. $t_1$ ($t_2$) represents low (high) water discharge.

yields insights into the conditions where subglacial sediment transport capacity could exhibit similar variability as in subaerial channels.

Channel types examined include subaerial channels (Figure 6 c), steady-state R-channels (Röthlisberger, 1972, Figure 6

d) and pipe-flow (i.e. R-channels of fixed size that do not adjust their size to discharge conditions; Figure 6 a). These formulations do not allow us to evaluate non-steady state R-channels presented above (Section 3.1, Figure 6 b). However, the formulations establish the scaling of these different channel typesc. The sediment transport capacity is calculated with a given water discharge and hydraulic gradient for three different sediment transport capacity formulas: Meyer-Peter Müller (MPM; Meyer-Peter and Müller, 1948), Engelund and Hansen (EH; Engelund and Hansen, 1967), and Bagnold (Bagnold, 1980);

additionally width-integrated shear stress is assessed, as used as proxy for sediment transport above.

Sediment discharge is given by the MPM, EH, and Bagnold formulations as

$$Q_s \propto w\tau^{3/2}, \quad Q_s \propto w\tau^{5/2}, \quad Q_s \propto w\left(\frac{\Omega}{w}\right)^{3/2} H^{-2/3}, \tag{11}$$

respectively, for sediment transport conditions well above the threshold of sediment motion.

We use the Darcy-Weisbach equation to evaluate shear stress

$$\Psi \propto f\frac{Q^2}{D_h S^2}, \tag{12}$$





with $\Psi = \Delta h/l$ the head gradient and friction factor $f$. The shear stress and stream power are, respectively,

$$\tau \propto f v^2 = f\left(\frac{Q}{S}\right)^2, \quad \Omega \propto \Psi Q. \tag{13}$$

As in Section 3.2, the subaerial channel is assumed to have a width $w$ much greater than its depth $H$, such that $D_h \approx 4H$, and to have a constant head gradient ($\Psi$) given by the topography. Further, it is assumed that its width can be approximated by a relation $w \propto Q^\alpha$ (Equation 9) with $\alpha \in [0,1]$. End members $\alpha = 0$ or $\alpha = 1$ correspond to a subaerial channel of constant width (a slot canyon) or depth (no natural equivalent), respectively. For a steady-state R-channel, it is assumed that $\Psi$ is constant (approximated by the gradient of the Shreve (1972) potential) and that $S$ adjusts in steady state with $\Psi$ and $Q$. Note that the R-channel model used above (Section 3.1), Equation 3 calculates $\Psi$ from the time-evolving $S_g$ via the Darcy-Weisbach Equation 4 and thus no Shreve approximation is then needed. Pipe flow-like conditions occur when an R-channel is subjected to rapid discharge variations such that the channel cannot adjust its size. In this case, it is assumed that the cross-sectional area $S$ is fixed, and $\Psi$ adjusts to the specified $Q$. For both steady-state R-channel and pipe flow, it is assumed that $D_h \propto S^{1/2}$.

With these assumptions, the Darcy-Weisbach equation (12) can be solved for the not-fixed quantity: $H$ for a subaerial channel, $S$ for a R-channel, and $\Psi$ for pipe-flow. Then, using equations (13), the shear stress and stream power can be calculated. These results are summarised in Table 2. The results of the 12 combinations of shear stress and sediment transport capacity are presented in Table 3 and also in Table S1 where the fractions in the exponents are approximately given by decimal numbers for ease of comparison.

**Table 2.** Relations for hydraulic variables for the three different channel types: subaerial channels, R-channel, and pipe-flow. Darcy-Weisbach equation is abbreviated with "D-W", and stream power with "Stream p.".

| Channel type | Fixed | Determined via D-W (Equation 12) | Additional relations | Shear stress $\tau \propto$ | Stream p. $\Omega \propto$ |
|---|---|---|---|---|---|
| Subaerial | $\Psi$ | $H \propto f^{1/3} Q^{2/3-2\alpha/3} \Psi^{-1/3}$ | $w \propto Q^\alpha$ $\quad$ $S = wH$ $\quad$ $D_h \propto H$ | $f^{1/3} Q^{2/3-2\alpha/3} \Psi^{2/3}$ | $Q\,\Psi$ |
| R-channel | $\Psi$ | $S \propto f^{2/5} Q^{4/5} \Psi^{-2/5}$ | $D_h \propto w \propto H \propto S^{1/2}$ | $f^{1/5} Q^{2/5} \Psi^{4/5}$ | $Q\,\Psi$ |
| Pipe | $S$ | $\Psi \propto f Q^2 S^{-5/2}$ | $D_h \propto w \propto H \propto S^{1/2}$ | $f Q^2 S^{-2}$ | $f Q^3 S^{-5/2}$ |

Table 3 shows the scaling of the proxy $w\tau$ as well as $Q_s$ of the MPM, EH, and Bagnold sediment transport formulas with respect to $Q$, $\Psi$ or $S$ for the three different channel types and $\alpha$ values. Remarkably, the Bagnold formula has a negative exponent for $f$ in all but the pipe-flow channel type. The total transport formula EH gives a slightly stronger dependence on all variables due to the larger exponent on $\tau$ of $\frac{5}{2}$ versus $\frac{3}{2}$ for the MPM (Equation 11). Albeit, the sediment transport response in pipe flow for the Bagnold case is close to EH. Conversely, the sediment transport proxies $w_g \tau_g$ and $w_f \tau_f$ used previously scale only as $Q^2$ for pipe-flow, whereas the sediment transport capacity scales at least with $Q^3$. The exponent for the width scaling $\alpha$ only impacts the relationship between sediment transport and water discharge in the EH relation in any meaningful





way. However, for the value of $\alpha$ around $\frac{1}{3}$ (a value appropriate for most streams), the exponent on $Q$ for EH is only slightly
greater than the other transport relations.

The sediment discharge capacity in the steady state R-channel scales very similarly to the subaerial channel case for all relations and virtually identically for the $\alpha = \frac{1}{3}$ case (Table 3). However, the head gradients $\Psi$ are likely higher for comparable $Q$ in an ice-sheet marginal or alpine glacier setting than in a subaerial channel (Alley et al., 1997). Thus, sediment transport capacity is likely higher in a steady-state R-channel.

**Table 3.** Sediment transport proxy ($w\tau$) and rates for the three considered different transport formulas: MPM (Meyer-Peter and Müller, 1948), EH (Engelund and Hansen, 1967), and Bagnold (Bagnold, 1980).

|  | Width $\times \tau$ | MPM | EH | Bagnold |
|---|---|---|---|---|
|  | $w\tau$ | $Q_s \propto w\tau^{3/2}$ | $Q_s \propto w\tau^{5/2}$ | $Q_s \propto w^{-1/2}\Omega^{3/2}H^{-2/3}$ |
| Subaerial | $f^{1/3}Q^{2/3+\alpha/3}\Psi^{2/3}$ | $f^{1/2}Q\Psi$ | $f^{5/6}Q^{5/3-2\alpha/3}\Psi^{5/3}$ | $f^{-2/9}Q^{19/18-\alpha/18}\Psi^{31/18}$ |
| R-channel | $f^{2/5}Q^{4/5}\Psi^{3/5}$ | $f^{1/2}Q\Psi$ | $f^{7/10}Q^{7/5}\Psi^{9/5}$ | $f^{-7/30}Q^{31/30}\Psi^{26/15}$ |
| Pipe | $fQ^2S^{-1}$ | $f^{3/2}Q^3S^{-5/2}$ | $f^{5/2}Q^5S^{-9/2}$ | $f^{3/2}Q^{9/2}S^{-14/3}$ |

R-channels rarely operate in a steady state with variations in water discharge, especially during severe rain or melt is too short for the channel to reach a steady state (Figures 2 and 3). In these cases with high water discharge variability, channels can behave more like a pipe of fixed cross-section. Here the cross-section responds to a characteristic discharge, but variations in water discharge deviate substantially from the flow conditions responsible for the channel size (i.e. diurnal water discharge variations in *ALPINE* case Figure 3; e.g. Gimbert et al., 2016). Table 3 shows that sediment transport in pipe-flow scales
much more severely with discharge; the exponent on $Q$ being between $3$ and $5$, compared to the other two channel types when that exponent is at most $\frac{5}{3}$. Thus fluctuations of discharge on short timescales (on the order of a day; Figure 4) have the potential to cause conditions with very high sediment transport capacities. Alternatively, at low water discharges these channels could become depressurized and transition to subaerial flow (Perolo et al., 2018). These sediment transport capacities variations are of far higher magnitude than those of subaerial channels (Alley et al., 1997). Note, however, that pipe flow
assumptions would cause sediment discharge capacity to covary with water discharge, as is not the case in normal R-channels (Figures 3 and 6 a and b). Covariance also occurs for steady state R-channels (Figure 6 d).

## 5 Discussion

### 5.1 Increased variability in sediment transport capacity in subglacial systems

The greater variations in shear stress in subglacial channels compared with subaerial channels that we present here may cause
even greater variations in sediment transport capacity in subglacial channels than our numerical results suggest (Figure 4). In sediment transport capacity relationships, such as in Meyer-Peter and Müller (1948) or Engelund and Hansen (1967), shear





stress is scaled to the power of $\frac{3}{2}$ or $\frac{5}{2}$, respectively (e.g. Section 4.3). The exponent greater than 1 magnifies sediment discharge variability beyond the variable sediment transport parameters described above (Figure 4; Table 3).

Greater variations in subglacial sediment transport capacity could cause a supply-limited regime at many glaciers due to
their high sediment transport capacity under assumed pressurized flow conditions (Alley et al., 1997). In subglacial systems, sediment's critical shear stress, or threshold at which sediment mobilization occurs, can be reached more frequently and across many water discharges, compared to subaerial systems (Figure 3). This result suggests sediment export here is especially sensitive to observed changes in water discharge variability (Lane and Nienow, 2019). Sediment exhaustion through repeatedly crossing the mobilization threshold may explain the stronger dependence of sediment discharge from the Greenland Ice Sheet
on the glaciers' basal shear stress, a proxy for bedrock erosion, rather than glacier melt (Overeem et al., 2017). More generally, the same process could also result in sediment discharge's strong dependence on sediment production from glacial sliding, itself dependent of basal shear stress of the glacier (Herman et al., 2015; Koppes et al., 2015).

While transport-limited states likely do not occur at many glaciers (e.g. Alley et al., 1997), abundant sediment could persist underneath some glaciers, potentially creating a transport-limited regime (e.g. Walter et al., 2014; Stevens et al., 2022; Delaney
and Anderson, 2022). At these glaciers, the great variability in subglacial sediment transport capacity may make it difficult to link sediment discharge to hydrology, especially when peak events occur (Cowan et al., 1988; Delaney et al., 2018; Lu et al., 2022). The results here suggest that different channel sizes with the same hydrology forcing can result in very different sediment transport capacities, making it challenging to establish the effects of individual extreme precipitation or melt events without establishing the antecedent state of the subglacial drainage system.

Catchments with reduced water discharge variability may experience less variability in sediment transport capacity, shown by the *ALPINE* and *ICESHEET* model outputs. Indeed, the more decoupled and sporadic relationship between model outputs and water discharge in *ALPINE* results from the relatively larger discharge variations on sub-daily to weekly timescales (Figure 3). This occurs as the subglacial channel's size evolves slowly compared to the variations in water discharge (Figure 6 b). High water discharge variability in *ALPINE* may cause width integrated shear stress to approach $Q^2$ in assuming pipe-flow
conditions where the water discharge varies largely compared to channel size (Figure 6; Section 4.3; c.f. Alley et al., 1997). Conversely, the reduced relative variability in water discharge in *ICESHEET* comes as a result of the larger catchment areas and longer travel times of the water (e.g. van As et al., 2017). Water discharge's stronger correlation with width-integrated shear stress in *ICESHEET* may result in the subglacial channel's size being closer to equilibrium from the smaller variations in water discharge (Figure 6 d). In this case, the exponent on water discharge for shear stress is likely substantially less than
$w\tau \propto Q^2$, but greater than $w\tau \propto Q^{\frac{4}{5}}$ that occurs in a steady-state R-channel (see Section 4.3). As a result, there is a stronger relationship between subglacial model outputs and water discharge in *ICESHEET* (Figure 3) and less variability model outputs in the *ICESHEET* case (Figure 4).

## 5.2    Interpreting sediment transport records from glacierized catchments with respect to water discharge

A sporadic relationship between subglacial sediment transport capacities and water discharge occurs due to hysteresis in sub-
glacial channel size (Figure 3). This hysteresis limits the use of water discharge as an indicator of sediment discharge capacity





in these systems. As a result, characteristics such as bankfull or effective water discharge that link geomorphic work to hydro-climatic conditions could have limited meaning in evaluating subglacial sediment transport (Wolman and Miller, 1960; Lenzi et al., 2006). A glacier's sediment transport capacity is impacted by the ice thickness controlling the channel closure rate and the glacier's surface slope, in addition to water discharge and sediment size (Figure 4, Section 3.1; Röthlisberger, 1972; Gimbert et al., 2016; Stevens et al., 2022; Walder and Fowler, 1994). This multitude of processes lies in contrast to many subaerial channels, where transport capacity typically responds to water discharge, sediment size, channel shape, and hydraulic gradient. The latter of these parameters can remain relatively stable over the years or longer (Section 3.2; e.g. Tucker and Slingerland, 1997). Strong correlations between water discharge and sediment export in glacier systems could indicate other processes, such as increased sediment access (Zhang et al., 2022). Furthermore, the observed correlation between water discharge and sediment discharge far downstream likely represents subaerial transport processes, especially with respect to bedload (Mancini et al., 2023).

Clockwise hysteresis loops between sediment concentration and water discharge in subaerial channels suggest that sediment availability is reduced over the event scale (Williams, 1989). These loops are often observed in glacierized catchments and used to suggest that access to subglacial sediment has been limited (e.g. Collins, 1979; Willis et al., 1996; Richards and Moore, 2003; Stott and Mount, 2007; Delaney et al., 2018). Results here suggest that clockwise hysteresis could also be expected in transport-limited regimes in many subglacial environments. Here, the smaller channel size on the rising limb would result in greater sediment transport capacity and sediment mobilization (Figure 6 b). The subsequently larger channel size would reduce sediment transport capacity for an equivalent water discharge on the falling limb, creating clockwise hysteresis.

The co-varying relationship between sediment transport capacity and water discharge in subaerial conditions could be noticeable $\sim 20\,\mathrm{km}$ downstream of the Leverett site. A strong correlation persists between sediment plume size and the Watson River's water discharge into the Kangerlussuaq fjord (Figure 6 c; Chu et al., 2009; McGrath et al., 2010). In contrast, in marine-terminating glacier catchments, a less consistent relationship may occur between water discharge or melt extent and sediment plume size (Chu et al., 2012; Tedstone and Arnold, 2012). Here, ocean water pressurizes subglacial channels at the ice front (e.g. How et al., 2017), so the observed reduced correlation could result from the inconsistent relationship between subglacial sediment transport capacity and water discharge (Figures 3 and 6 b).

Water discharge measurements at sub-daily timescales could severely limit the ability of a subglacial sediment transport model to capture specific events. The decreased model output variability beyond 1–5 days of smoothing could result in water discharge appearing to be in equilibrium with the subglacial channel when they are in fact not (Figure 4). Over this period, water discharge variations could lead to a stronger relationship with sediment transport capacity (Figure 6 d and Section 4.3). Particularly, if water discharge evolves slowly with respect to the subglacial channel size, then it will covary with sediment transport capacity (Figure 6 d). Such a strong relationship would not represent the impact of actual shorter-term fluctuations in water discharge.



### 5.3 Experiment limitations

Several aspects of the models and experiment design make the comparison of the subaerial and subglacial systems difficult.

The lumped nature of the models means that they operate independently of the upstream drainage network. Additionally, we omit analysis of suspended sediment discharge records due to their dependence on sediment supply in capturing variations in sediment discharge, in addition to sediment transport capacity (e.g. Delaney et al., 2019). In reality, processes such as sediment access are usually important in controlling sediment export in glacierized catchments (e.g. Herman et al., 2015; Vergara et al., 2022). Furthermore, meltwater can be distributed to flow through several adjacent conduits impacting the sediment transport

capacity in each (e.g. Werder et al., 2013; Hewitt and Creyts, 2019; Delaney et al., 2023). The lumped models used here isolate the relationship between water discharge and sediment transport capacity in subglacial and subaerial systems. However, they neglect more complex, yet important, spatially distributed processes.

Different hydraulic gradients control the velocity and sediment transport capacity in the subglacial and subaerial cases (Section 3.1 and 3.2). The subglacial one is generally much steeper (Alley et al., 1997). This results in the shear stresses

and width-integrated shear stresses across the channel bed that controls sediment transport capacity being much greater in subglacial channels (Figure 2). The parameters' range tested in Section 4.2 covers a likely span of viable shear stresses in both subglacial and subaerial channels. The greater sediment transport capacity in the subglacial channels implies that sediment grain size underneath subglacial channels is likely larger than subaerial counterparts. Note, however, that the water velocities are similar in both channel types.

The channel width and size variations in subglacial channels presented here also occur in subaerial channels in response to water discharge (Phillips and Jerolmack, 2016), likely over decadal timescales or in response to individual extreme events (e.g. Slater and Singer, 2013; Dean and Schmidt, 2013). These timescales are likely considerably longer than the continuous changes to the subglacial channel's response time of days. Furthermore, observations from subaerial systems can show a co-varying relationship between water discharge and sediment transport in time (e.g. Schmidt and Morche, 2006; Pitlick et al.,

2021). Even so, if channel width changes occur in subaerial channels, then the values of $\alpha$ in Equation 9 could change in time. $\alpha$ must be less than 1, and even this would cause the exponent on $Q$ in the $w_g \tau_g$ relationships to remain substantially below that of pipeflow (Section 4.3, Table 2 and 3)).

Width-integrated shear stress is examined in the two numerical experiments as it does not have a dependence on grain size, unlike sediment transport relationships (Meyer-Peter and Müller, 1948, Section 4.3). This makes comparison between

the two systems simpler. Preferential transport of smaller sediment clasts and the input of upstream sediment impact sediment transport capacity as grain size evolves (e.g. Gomez, 1983). These processes are only beginning to be evaluated underneath glaciers (Aitken et al., 2024), but can be important in subaerial systems. Subglacial sorting processes could be an additional source of variability in sediment export from glaciers with respect to water discharge, especially for bedload transport.





## 6 Conclusions

The sediment transport capacity of both subglacial and subaerial channels is driven by its width and the shear stress exerted by the flowing water, which is proportional to flow velocity squared. Subaerial channels immediately alter both their width and water velocity in response to changing water discharge. In contrast, pressurized subglacial channels largely accommodate rapidly changing water discharge by altering water velocity, while their size only responds to changes in discharge over longer timescales, typically, several days. Thus, sediment transport capacity is more sensitive to changes in water discharge in

subglacial channels compared to subaerial ones.

The manuscript's first objective is to establish if water discharge covaries with subglacial sediment transport capacity. In subglacial channels, the timing of peak water velocity and sediment transport capacity occurs before peak water discharge during a discharge event, due to evolving channel size. In subaerial channels, the timing of peak sediment transport capacity and water discharge coincide. Results here suggest that, even in a transport-limited subglacial system, an incoherent relationship

between water and subglacial sediment discharge could be expected in most channels. This incoherent relationship presents a challenge in linking hydro-climatic conditions or events to sediment export from glaciers. Water discharge and sediment transport capacity covariance could be possible when water discharge varies at a slower rate than subglacial channel size. In natural environments, this appears to rarely be the case.

The manuscript's second objective aims to evaluate the relative variability in sediment transport capacity between subglacial

and subaerial channels. Results demonstrate that sediment transport capacity variability is higher in pressurized subglacial channels that are out of equilibrium with water discharge. Increased variability occurs even in environments where the water discharge has relatively small diurnal variations in the *ICESHEET* case. Yet, elevated variability is especially strong with greater water discharge variations in the *ALPINE* case. Further evaluation is needed to establish the role of high variability in sediment transport capacity in evaluating hydro-climatic signals from sediment records. Greater variability in sediment

transport capacity may also lead to sediment exhaustion by subglacial water repeatedly crossing the mobilization threshold across a range of water discharges.

Few observations of the subglacial environment hinder our ability to quantify processes such as the shape of subglacial channels and the response time of subglacial channels to water discharge variations. The poor constraints on subglacial sediment size and sorting processes make it difficult to link shear stress to sediment transport capacity. Further quantifying these processes

will help to better inform the response of sediment transport capacity to water discharge forcing in subglacial environments.

This study calls for the explicit consideration of evolving channel size when examining the relationship between sediment transport and hydro-climatic conditions from glacierized catchments, especially ones with high water discharge variability.

*Code and data availability.* Code, with links to the data, can be found at https://bitbucket.org/IanDelaney/xsection/src/master/. Data from Leverett glacier have been previously published in Tedstone et al. (2013). Data from Fieschergletscher have been previously published in

Delaney et al. (2024). The code will be uploaded to a permanent repository, with FAIR principles pending acceptance of the paper.



*Author contributions.* I.D. designed the study, developed, and implemented the model and experiments, and wrote the manuscript. A.J.T. helped interpret findings from Leverett Glacier and guided experiment design. M.A.W. and D.F. provided data from Fieschergletscher and contributed to analysis. All authors provided key inputs in writing and editing the manuscript.

*Competing interests.* The one or more of the (co-) authors is a member of The Cryosphere editorial board and is not involved in the review
process.

*Acknowledgements.* SNSF Project No. PZ00P2_202024 provided funding for I. Delaney. A. Tedstone acknowledges funding from the European Research Council, award 818994 – CASSANDRA. G. King and M. J. Gevers provided useful comments on a previous version of this manuscript. We benefited from fruitful discussions with J. Irving. We thank two reviewers for constructive and critical comments on a previous version of the manuscript.



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
