# Peer review of "Sediment transport capacity response to variations in water discharge in pressurized subglacial channels"

_EGUsphere, 2024_

## Referee Comment (RC1)

Subglacial and subaerial fluvial sediment transport capacity respond differently to water discharge variations

Author(s): Delaney and others

**General comments**
In my view the paper is interesting and relevant, and hence worthy of publication. I have a couple of general observations, followed by a list of minor suggestions.

My general observation relates to the structure of the main findings reported by the manuscript. In reading the Results and Discussion I was sometimes lost as to what I was learning that was new. I think this was mainly because the structure lacked a clear and logical progression from one general finding through progressively more specific findings. This may well stem an initial imperfect expression of the objectives. Here, most of what is novel seems to fgall under Objective 2 and not under Objective 1. Indeed, isn't Objective 1 ('to establish whether sub-seasonal water discharge can co-vary with sediment transport capacity in subglacial systems') already established beyond reasonable doubt? To me, and I believe the introduction to this manuscript, it clearly 'can'… (note the objective refers to 'can it', and not to 'does it' [to which the answer is also almost certainly 'yes' but perhaps with a little more room for maneuver]). I would recast the objectives to reflect better the new material presented in this manuscript (closer to objective 2 in fact, which I feel could readily be sub-divided). Having considered this, I feel the progression of the Results could be far more accessible to the reader and allow the manuscript to focus on a few really key points (which I think are currently slightly lost in the density of the presentation).

Interpretation is presented within each Results section (e.g., see 'due to… on L174). This continues throughout Results, and I would separate all of the explanation and hence interpretation out form the results.

There are some very important points in Results, which I would attach some key data to and make more explicit here and in the Conclusion and Abstract. I'm thinking for example of associating the (increased) hysteresis as a function of discharge with a headline % change in sediment transport capacity. E.g., "thus, a more variable, but typical, Alpine Q record can transport up to *% greater sediment per cumec than a less variable ice sheet-type Q" and/or "the greater variability in Q from Alpine than from ice sheet style Q can account by an offset of peak sediment discharge from peak meltwater discharge by up to * % of the cycle involved". At present, I feel the manuscript lacks this incisive illustration to bring the scale of the results home to the reader.

Less importantly, but something I'd like to check, is that the fundamental conclusions are applicable more generally. Since they derive from modelling based on two 'type' datasets, there may be a possibility that not all key findings hold for certain other variations of subglacial discharge pattern. I have every reason to believe they do, but I feel the manuscript should report that this has been done more widely and the key results/patterns still hold. I don't think there's any need to include any such results into a revised manuscript – but I would just make the statement that while, for simplicity of illustration, two typical but contrasting datasets are explored here, the model has been run with other data sets and the key do hold more generally. Incidentally, I would also include an explicit disclaimer early on that the analysis only relates to pressurized subglacial flow and not to open-channel flow prominent approaching the terminus of many valley glaciers etc.

**Specific comments**

I

| Line/Location | Comment/Suggestion |
|---|---|
| Title | The title states the obvious from prior research and I would recast it to reflect more accurately the new and original contribution of this manuscript. Also, if the authors elect to not change the title I would change the existing wording to 'transport capacities respond differently'. Grammatically, this is not clear given the two capacities noted earlier in the title, but stylistically I think 'capacities respond' reads better than does 'capacity respond'. (I would change the whole thing anyway… Why not start by considering 'Sediment transport capacity response to variations in water discharge in pressurized subglacial channels') |
| 3 | 'full' subglacial channels? |
| 5 | There is no need to mention 'over days' here (indeed, it is misleading anyway without quantification of how much change as a function of time). I would just leave it as changes slowly (or slowly relatively to variations in water Q). |
| 7 | 'Sheet' is missing (the authors really should have picked up on this obvious typo) |
| 8 | This hysteresis causes (no need to qualify the type of hysteresis since the use of 'this' serves the purpose) |
| 64 | I think there is earlier use of transport-limited (in which case the definition should be presented there, and not here) |
| 75-76 and 80-81 | I'd remove summary of the results of this study from the Introduction. They are not yet established. |
| 70-71 | In phrasing objective 1 (in more than one place in the manuscript), shouldn't the dependent variable come first? Thus, I would write it as 'to establish whether sediment transport capacity can co-vary with water discharge in subglacial systems'. |
| 79 | Given the apparent relevance of the Alley et al. study, I feel it's key findings should be presented in detail here to provide more direction to the specific objectives of this study. |
| 159 | The parameter values are not purely random (but random within certain sensible ranges, no?). |
| 171- | I get a little muddled right at the start here – not helped by the sub-heading, which I would recast as "Influence of subglacial channel size on the timing and variability of…". The section (and others following) also combines results with explanation (i.e., interpretation) which I feel also adds unnecessarily to the difficulty in following the progression. I would separate results from interpretation. |
| 172 | This narrative jumps straight in with the ambition to 'quantify the sources of increased variability…' but the nature of that increased variability has not yet be established or illustrated. This really needs building up more progressively – bringing the reader through with the first-order model results. Then go on to explore more and more detailed influences and relationships. Most of this is, in fact, in Figure 2 – but the reader is not guided through the fundamental relationships here. |

| | |
|---|---|
| 173 | Change rather than evolution? (elsewhere too) |
| Fig 2 caption | Is it really an arbitrary y axis scale? If it is truly arbitrary, does it even need a scale range? |
| 179-183 | Is this not already established? If so, it should be presented clearly in the Introduction and developed as necessary here. It is also interpretation. |
| Fig 3 (and Fig 4) | Shouldn't the dependent variable be plotted on y and the independent variable on x? I for one at least think this way around and therefore find these plots a little confusing. |
| | Is there a need for the axis labels to be at an angle rather than parallel to the axis? If not, I would reorientate them parallel in all cases. |
| 193-4 | This is interpretation |
| 195 | The reference to 'increased variability' needs to make it clear what property is being referred to. Like the previous sub-heading, this sub-heading is a statement of findings before those results have been presented. I would avoid this and present it as the less assuming: 'Influence of…' |
| 205-6 | Interpretation in Results |
| 210-11 | Interpretation in Results |
| 220 | Typo (subglacial) |
| Section 4.3 | A lot of this reads more like methods and background modelling operation and/or equation terms than Results of the modelling. If it is all results then the key findings are difficult to pull out from the density of the presentation – perhaps some of it could be omitted or moved to supplementary? |
| Fig 5 | I find the figure too small. If larger the font would be easier to read and the y axes could have the property written out rather than represented by symbols. I'm sure there is no need for it to be this small. |
| 232 | Typo? ('typesc') |
| 235 | The text: '…as used as proxy for…') doesn't make sense (or at least could be worded more clearly). |
| 284 (maybe elsewhere too) | The wording should be "greater …. than" and not 'greater… compared with…). |
| 297 | "…itself dependent of…" doesn't make sense to me |
| 298 | I don't believe there is sufficient evidence to make the claim: 'While transport limited states likely do not occur at many glaciers'. In my own experience most glacier do have at least parts of the bed that are sediment-rich and hence have the capacity to be transport limited. |
| 302-4 | I would try to pull some key quantification out here and present it as a headline figure to illustrate the magnitude of the effect. |
| 319 | I'm not sure 'sporadic' is the right word here. Please check you mean this. |
| 321-22 | I see the point being made here but it's not black or white. Surely there is some control exerted by discharge; it's just that it is complicated by the additional forces in the subglacial scenario. The way this is presented here may be technically ok but the reader could be forgiven for coming away with the feeling that there is no solid relationship between Q and sediment transport capacity is these cases – which would not be accurate. I feel it would be particularly helpful if the effect could be quantified in some generalized or headline way in terms of the likely (or maximum if you |

| | |
|---|---|
| | prefer) effect on a typical alpine and/or ice sheet hydrograph. That figure could then also go into the Conclusions and Abstract. |
| 394 | Surely the relationship can be characterized a little more usefully than simply to state that it is 'incoherent'. This also relates to my point above relating to lines 321-22. Can this be characterized accurately or systematically so the reader has a little more to learn in terms of the nature and magnitude of the effects? |
| 407-10 | It would not do any harm here to point to advances in subglacial sediment tracking, which speaks to this issue. I am well familiar with Jenkins' work – but there may well be others too. |

---

## Referee Comment (RC2)

**Subglacial and subaerial fluvial sediment transport capacity respond differently to water discharge variations**

I. Delaney, A. J. Tedstone, M. A. Werder and D. Farinotti

The Cryosphere 2024-2580

*Referee's report*

The paper is concerned with the nature of sediment transport in subglacial channels, and its difference to that in sub-aerial channels. There is a three page introduction which references lots of papers, but I didn't really get much from this. The two models for subglacial and sub-aerial channels are presented in section 3.

The subglacial model is a lumped one, in which the effective pressure controlling channel closure is related to the hydraulic head drop $\Delta h$, which thus makes its appearance in both the closure equation (3) and the momentum equation (or force balance), equation (4). I am a bit suspicious about this, as also discussed in the small item (eq. 3) below. I am thinking of Nye's 1976 model as a framework here, although with an added source term in the mass conservation equation to allow for surface meltwater input, subglacial tributaries, and the like.

Now the time scale for changes in water flow is much faster than that due to channel closure, with the consequence that the mass flow equation is effectively at steady state. Further, the mass source term due to wall melting is very small *in the mass flow equation*, so integration of this equation just gives the water discharge in terms of the inflow. Then $v = Q/S$, so the force balance in (4) determines head loss as a function of $S$, and then (3) is a single ordinary differential equation for $S$.

The trouble is, that while the first term on the right hand side of (3) makes sense to me, the second does not, and the reason for this is that the closure term involves the effective pressure and thus the hydraulic head, but not its gradient, so I am a bit sceptical about this, because the structure of the Nye model seems to have been altered. You might say that the distinction disappears in the lumping, but it seems to me that if you have an upstream head $h_-$ and a downstream head $h_+$, the first term on the right of (3) will involve the difference, but the second will involve the average, and there are *two* independent quantities involved. I suppose you can get around this by saying, oh, the outlet head or more accurately the effective pressure is zero, and perhaps that is what is done. But that is a bit disingenuous, because the outflow becomes open channel flow somewhere upstream of the actual mouth of the stream.

Now I went and looked up this Werder 2010 paper, and you can see in its equation (4) the same distinction I am making here. I suppose my overall view is that (I think) the Clarke paper intended to use the flow elements of resistors, etc., as ingredients of a larger scale flow path, and this lumping of a whole channel is an extremely coarse thing to do, and looks a bit like avoiding confronting the spatially-dependent physics simply because it's too hard. So I think you can rescue this aspect of the model presentation, but a bit more exposition would help, in particular the figure 2

or equivalent of the 2010 paper would help. And the model should come with some caveats, e. g., you could say that it is a simplistic and possibly unreliable model. To be fair, this is done in section 5.3, but that is too late.

One of the comments made is that because the sub-aerial model has an algebraic relation between width and discharge (equation 9), the relationship between velocity and discharge is also algebraic; but this is simply a choice of the model. In reality, the width of a channel will also evolve over (long) time through processes of bank erosion, and the use of equation 9 properly only involves a long term time average of discharge, so it is misleading to use it in a time-specific way. Actually, this is admitted at line 370, along with other limitations in the models.

Two kinds of numerical experiment are described in section 3.3, and the results of these presented in section 4. Presumably the input to the model runs is the (time-varying) water discharge, but that seems not to be stated in 3.3, which I would expect, although it is stated in the caption to figure 2.

The second half of the paper characterises the results of these experiments and there is an extended discussion. There is no doubt the authors have put a lot of effort into this, but my interest waned at this point.

In summary, this is one of those difficult papers to judge, because the effort involved is quite substantial and honestly applied, but the whole philosophy of the approach is, in my view, misguided. This is a coxless boat crew who are rowing up a backwater without a rudder, and have lost the main direction of the stream.

I can perhaps come to a conclusion if I imagine future researchers reading and referencing this paper. They might want to say, "Delaney *et al.* (2025) showed that . . . "; but they didn't. What they did was take two models of subglacial and sub-aerial stream and sediment transport, and study their consequences in response to measured hydrographs. So the conclusions are only as good as the models. But there are big conceptual holes in the models themselves; for example, Meyer-Peter/Müller is only one of several such relationships, and it itself is of its nature a steady state result, and its application subglacially has to be a shot in the dark. The subglacial model assumes particular channel shape, and furthermore, the choice of a lumped parameter simplification, while it can be defended to some extent, clearly steps away from an effort to provide the best possible description. The only way some kind of rescue act could be performed would be if the whole ethos were changed; if for example, outlet discharge and sediment transport were both measured, and for example, plotted in a phase plane where circular paths were obtained; that would then suggest hysteresis, and a model, even such a diaphanous one as this, would have some merit. But the model must serve the data, and not the other way round. I think this paper must be rejected for that reason.

Some smaller points (line numbers or equation numbers in parenthesis):

(33): I don't know what 'Following mass conservation' has to do with this.

(99-100): upon that of Clarke.

(102): presumably $h_{\text{ice}}$ is the thickness of the ice, but the syntax is not clear, it could be the depth of the channel.

(eq. 3): it's probably in the Werder 2010 paper, but I'd like to see an extra comment about where this $\dfrac{\Delta h}{2}$ term comes from. Since the bracket represents $N = p_i - p_w$, the pointwise form of this second term would be $\dfrac{p_w}{\rho_w g}$, so it's not obvious to me where the $\dfrac{\Delta h}{2}$ comes from. Also in this light, it is worth elaborating the confusing term 'hydraulic head drop change' and defining what $\Delta h$ is in terms of $p_w$. Actually, the more I think about this, the more suspicious I become. And in fact, at least when you're dealing with jökulhlaups, the discharge doesn't vary much along the length of the channel, and the consequence of that is that neither does the effective pressure. Of course, that may not apply here but I think the same principle applies. So maybe I am promoting this to a more substantive issue (as indeed now further discussed earlier).

(114): I had no idea what the central angle referred to, but evidently it is the angle subtended at the centre of the circular arc of the channel upper boundary; and then $\beta = 2\alpha$, where $\alpha$ is the contact angle at the channel edge, which seems a more natural quantity to use. Also I checked the algebra of equation (5), and got this formula, but with the factor of two in the numerator, not the denominator, assuming hydraulic diameter is twice hydraulic radius, the latter of which is area/perimeter, so please check this. Incidentally, I did also then check equation (7) and I agree with that, so I do think there is an error in (5).

---

## Author Comment (AC1)

Dear Editor,

We thank Reviewer 1 for their detailed and constructive comments. Their work, we believe, has provided feedback that has greatly improved the manuscript.

The reviewer's comments are in bold, our response is in italics and quotations from the new text are in normal font.

Best regards,

Ian Delaney on behalf of all authors

**General Comments**

- **In my view the paper is interesting and relevant, and hence worthy of publication. I have a couple of general observations, followed by a list of minor suggestions. My general observation relates to the structure of the main findings reported by the manuscript. In reading the Results and Discussion I was sometimes lost as to what I was learning that was new. I think this was mainly because the structure lacked a clear and logical progression from one general finding through progressively more specific findings. This may well stem an initial imperfect expression of the objectives.**

  *We thank the Reviewer for their positive assessment of our work and acknowledge the structural and organizational issues they have highlighted.*

- **Here, most of what is novel seems to fall under Objective 2 and not under Objective 1. Indeed, isn't Objective 1 ('to establish whether sub-seasonal water discharge can co-vary with sediment transport capacity in subglacial systems') already established beyond reasonable doubt? To me, and I believe the introduction to this manuscript, it clearly 'can'. . . (note the objective refers to 'can it', and not to 'does it' [to which the answer is also almost certainly 'yes' but perhaps with a little more room for maneuver]). I would recast the objectives to reflect better the new material presented in this manuscript (closer to objective 2 in fact, which I feel could readily be sub-divided). Having considered this, I feel the progression of the Results could be far more accessible to the reader and allow the manuscript to focus on a few really key points (which I think are currently slightly lost in the density of the presentation). Interpretation is presented within each Results section (e.g., see 'due to. . . on L174). This continues throughout Results, and I would separate all of the explanation and hence interpretation out form the results.**

  *We greatly appreciate this comment. After careful consideration, we have chosen to rephrase Objective One as:* "to establish the hydrological conditions under which sediment transport capacity co-varies with water discharge in subglacial systems."

  *In our view, this objective directs attention to the differences between the ALPINE and ICESHEET scenarios. It is also practical for evaluating the effects of water discharge smoothing, as discussed in the next point.*

  *We acknowledge the importance of removing interpretation from the results section and have made the necessary revisions.*

- **There are some very important points in Results, which I would attach some key data to and make more explicit here and in the Conclusion and Abstract. I'm thinking for**

**example of associating the (increased) hysteresis as a function of discharge with a headline % change in sediment transport capacity. E.g., "thus, a more variable, but typical, Alpine Q record can transport up to \*% greater sediment per cumec than a less variable ice sheet-type Q" and/or "the greater variability in Q from Alpine than from ice sheet style Q can account by an offset of peak sediment discharge from peak meltwater discharge by up to \* % of the cycle involved". At present, I feel the manuscript lacks this incisive illustration to bring the scale of the results home to the reader.**

*This is an excellent comment, and we have worked to provide a clearer explanation of hysteresis.*

*One challenge with the approach recommended by the reviewer is that sediment transport capacity scales highly non-linearly with water discharge, particularly in subglacial systems, as demonstrated in Section 4.3. Consequently, findings across different hydrographs would not be directly comparable, even if they were scaled using a discharge quantity.*

*Instead, we have added several paragraphs to Section 4.1 discussing the role of water discharge variability in hysteresis.*

*Here the aim is to evaluate the effects of smoothing hydrographs, yielding insights in to the hydrological conditions where discharge and sediment transport capacity can co-vary. Note that the smoothed hydrographs used for the analysis here are presented in the supplement. A version of the new paragraphs is as follows.*

[Figure]

Figure 1: Spearman rank correlation between water discharge over a smoothed period and sediment transport characteristics (a) water velocity, (b) shear stress, (c) width-integrated shear stress. Higher correlation scores mean reduced hysteresis and approach subaerial behavior.

Water discharge and sediment transport characteristics correlate better in *ICESHEET* with lower diurnal variations in water discharge compared to the *ALPINE*. To evaluate the impact of water discharge variability, we smoothed the two hydrographs across different periods and evaluated the Spearman rank correlation between sediment transport characteristics and water discharge. We assume that a higher rank correlation indicates a reduced amount of hysteresis and behavior more similar subaerial channels (Figure 3).

Smoothing water discharge over periods longer than one day causes a substantial increase in rank correlation (Figure 5). This increase occurs as diurnal variations are removed (Figures S1-S14). Correlations in sediment transport characteristics in *ICESHEET* remain higher than *ALPINE*, which has more water discharge variations even when smoothed compared to *ICESHEET* (Figure S1-S14).

- **Less importantly, but something I'd like to check, is that the fundamental conclusions are applicable more generally. Since they derive from modelling based on two 'type' datasets, there may be a possibility that not all key findings hold for certain other variations of subglacial discharge pattern. I have every reason to believe they do, but I feel the manuscript should report that this has been done more widely and the key results/patterns still hold. I don't think there's any need to include any such results into a revised manuscript – but I would just make the statement that while, for simplicity of illustration, two typical but contrasting datasets are explored here, the model has been run with other data sets and the key do hold more generally.**

  *We thank the reviewer for this comment as it, in part, encouraged us to add the additional section above with the water discharge smoothing. The other hydrological scenario we are aware of not explicitly considered here is Antarctica, where little surface melt occurs. Thus, we expect variations in water discharge to be minimal. The following sentence has been added to the conclusion:* "Water discharge and sediment transport capacity covariance could be possible when water discharge varies at a slower rate than subglacial channel size, such as in Antarctica with minimal surface melt input."

- **Incidentally, I would also include an explicit disclaimer early on that the analysis only relates to pressurized subglacial flow and not to open-channel flow prominent approaching the terminus of many valley glaciers etc.**

  *We will add the following sentence to the model implementation section:* "We also note that the subglacial results apply to pressurized subglacial channels, not depressurized one that can occur below glacier termini (Perolo et al., 2018)"

**Specific Comments**

- **Title:** The title states the obvious from prior research and I would recast it to reflect more accurately the new and original contribution of this manuscript. Also, if the authors elect to not change the title I would change the existing wording to 'transport capacities respond differently'. Grammatically, this is not clear given the two capacities noted earlier in the title, but stylistically I think 'capacities respond' reads better than does 'capacity respond'. (I would change the whole thing anyway… Why not start by considering 'Sediment transport capacity response to variations in water discharge in pressurized subglacial channels')

  *Excellent suggestion, the title follows the reviewer's suggestion:* Sediment transport capacity response to variations in water discharge in pressurized subglacial channels

- **3:** 'full' subglacial channels?

  *Done.*

- **5:** There is no need to mention 'over days' here (indeed, it is misleading anyway without quantification of how much change as a function of time). I would just leave it as changes slowly (or slowly relatively to variations in water Q).

  *Done.*

- **7:** 'Sheet' is missing (the authors really should have picked up on this obvious typo)

  *Done.*

- **8:** This hysteresis causes (no need to qualify the type of hysteresis since the use of 'this' serves the purpose)

  *Done.*

- **64:** I think there is earlier use of transport-limited (in which case the definition should be presented there, and not here)

  *We thank the reviewer for the careful read. The definition was presented in Line 28, and thus we will remove the definition here.*

- **75-76 and 80-81** I'd remove summary of the results of this study from the Introduction. They are not yet established.

  *These lines will be removed.*

- **70-71** In phrasing objective 1 (in more than one place in the manuscript), shouldn't the dependent variable come first? Thus, I would write it as 'to establish whether sediment transport capacity can co-vary with water discharge in subglacial systems'.

  *Given the discussion above, the text will read:* "to establish the hydrological conditions where sediment transport capacity co-varies with water discharge in subglacial systems"

- **79** Given the apparent relevance of the Alley et al. study, I feel it's key findings should be presented in detail here to provide more direction to the specific objectives of this study.

  *We believe that this is discussed in Lines 51–54 in the introduction.*

- **159** The parameter values are not purely random (but random within certain sensible ranges, no?).

  *The text will now read:* "with random parameter values in a range"

- **171-** I get a little muddled right at the start here – not helped by the sub-heading, which I would recast as "Influence of subglacial channel size on the timing and variability of...". The section (and others following) also combines results with explanation (i.e., interpretation) which I feel also adds unnecessarily to the difficulty in following the progression. I would separate results from interpretation.

  *This section will be retitled:* "Timing of subaerial and subglacial sediment transport capacity variations." *This discusses the role of water discharge on hysteresis in water velocity or sediment transport capacity. Thus dealing with Objective One of the manuscript. We have also noted the comment about results and interpretation. These have been removed from the manuscript.*

- **172** This narrative jumps straight in with the ambition to 'quantify the sources of increased variability...' but the nature of that increased variability has not yet be established or illustrated. This really needs building up more progressively – bringing the reader through with the first-order model results. Then go on to explore more and more detailed influences and relationships. Most of this is, in fact, in Figure 2 – but the reader is not guided through the fundamental relationships here.

  *Excellent comment. The first sentence of the section will read:* "The first numerical experiment aims to quantify the timing and covariance of the subglacial model outputs with respect to water discharge in response to different seasonal evolutions and peaks"

- **173** Change rather than evolution? (elsewhere too)

  *We appreciate the comment. We believe that given the nature of subglacial channels, as described in Equation 3, evolution is a more accurate description. To us, "change" could include a modification for any number of reasons. However, "evolution" connotates the dependence on antecedent conditions which we believe to be more accurate in this case.*

- **Fig 2 caption** Is it really an arbitrary y axis scale? If it is truly arbitrary, does it even need a scale range?

*We believe that a description of the insets' axis is needed as these lines have the same axis in the main plots. However, their very different values mean that it makes sense to adjust values to examine their changes during the extreme melt event. The text will be modified slightly to read:* "different y-axis ranges for subaerial and subglacial values."

- **179-183** Is this not already established? If so, it should be presented clearly in the Intro-duction and developed as necessary here. It is also interpretation.

  *To the best of our knowledge this is the first time that this finding is being presented. We would welcome any citations or references that suggest otherwise. We believe that the content in this paragraph describes the response to water discharge in Figure 2. We agree that the reference to the "Methods" section, gave the impression of the interpretation. The paragraph will be adjusted slightly to read:*

  Peaks in subaerial model outputs occur coincident with peaks in water discharge (Fig-ure 2). In the subglacial channel, peaks in model outputs generally occur when water discharge increases, but before the maximum water discharge. As the water discharge stabilizes at its peak, channel growth continues (Figure 2 a, e), causing water velocity and other model outputs to decrease from their peak values (Figure 2 b-d, f-h). Subglacial sed-iment transport capacity is greatest on the hydrograph's rising limb, relative to the falling limb, creating a hysteresis effect."

- **Fig 3 (and Fig 4)** Shouldn't the dependent variable be plotted on y and the independent variable on x? I for one at least think this way around and therefore find these plots a little confusing. Is there a need for the axis labels to be at an angle rather than parallel to the axis? If not, I would reorientate them parallel in all cases.

  *The axes will be changed in both figures*

- **193-4** This is interpretation.

  *This sentence will be removed.*

- **195** The reference to 'increased variability' needs to make it clear what property is being referred to. Like the previous sub-heading, this sub-heading is a statement of findings before those results have been presented. I would avoid this and present it as the less assuming: 'Influence of...'

  *The section head will be modified to read:* Variability in sediment transport capacity across a range of channel shapes, slopes, and friction values

- **205-6** Interpretation in Results

  *Removed.*

- **210-11** Interpretation in Results

  *Sentence will be changed to:*" Smaller values of channel factor $\beta$, creating low and broad channels where the channel width grows more quickly in response to water discharge increases as compared to a semi-circular channel with $\beta = \pi$ (Equation 4)."

- **220 Typo (subglacial)**

  *Fixed.*

- **Section 4.3** A lot of this reads more like methods and background modelling operation and/or equation terms than Results of the modelling. If it is all results then the key findings are difficult to pull out from the density of the presentation – perhaps some of it could be omitted or moved to supplementary?

*We have considered the comments here, and with the editor's support, think it fits best in the manuscript. However, we have moved it to the discussion to set up the end member cases of pipe flow and steady-state R-channels. In a previous version of the manuscript, it was included in the supplementary material. However, after those reviews, we believe that some misunderstandings emerged. We believe that Tables 2 and 3 in the section are valuable results in evaluating the sensitivity of sediment transport capacity to water discharge under different assumptions.*

*We will modify the introductory text to read*: "The numerical experiments above consider the size evolution of subglacial channels and demonstrate that for these hydrographs sub-glacial sediment transport variability is greater than its subaerial counterpart (Section 4.3). Additionally, results demonstrate the impact of water discharge variability on sediment transport capacity. Here, we compare the sediment transport behavior of different channel types as they respond to water discharge, channel shape, and hydraulic gradient."

- **Fig 5** I find the figure too small. If larger the font would be easier to read and the y axes could have the property written out rather than represented by symbols. I'm sure there is no need for it to be this small.

  *We will increase the size of the axis labels.*

- **232** Typo? ('typesc') *Removed.*

- **235** The text: '...as used as proxy for...') doesn't make sense (or at least could be worded more clearly).

  *Text will read: "additionally width-integrated shear stress is assessed, instead of sediment transport, as above."*

- **284** (maybe elsewhere too) The wording should be "greater .... than" and not 'greater... compared with...').

  *Done.*

- **297** "...itself dependent of..." doesn't make sense to me

  *Text now reads: "which is dependent"*

- **298** I don't believe there is sufficient evidence to make the claim: 'While transport limited states likely do not occur at many glaciers'. In my own experience most glacier do have at least parts of the bed that are sediment- rich and hence have the capacity to be transport limited.

  *We agree with the reviewer about the lack of data. This sentence will be removed.*

- **302-4** I would try to pull some key quantification out here and present it as a headline figure to illustrate the magnitude of the effect.

- **319** I'm not sure 'sporadic' is the right word here. Please check you mean this.

  *We will replace "sporadic" with "variable".*

- **321-22** I see the point being made here but it's not black or white. Surely there is some control exerted by discharge; it's just that it is complicated by the additional forces in the subglacial scenario. The way this is presented here may be technically ok but the reader could be forgiven for coming away with the feeling that there is no solid relationship between Q and sediment transport capacity is these cases – which would not be accurate. I feel it would be particularly helpful if the effect could be quantified in some generalized or headline way in terms of the likely (or maximum if you prefer) effect on a typical alpine

and/or ice sheet hydrograph. That figure could then also go into the Conclusions and Abstract.

*The sentence will read:* "This hysteresis can limit water discharge as an indicator of sediment discharge capacity in these systems, especially when water discharge is highly variable and out of equilibrium with subglacial channel size."

- **394** Surely the relationship can be characterized a little more usefully than simply to state that it is 'incoherent'. This also relates to my point above relating to lines 321-22. Can this be characterized accurately or systematically so the reader has a little more to learn in terms of the nature and magnitude of the effects?

*This paragraph will read:* "The manuscript's first objective is to establish the conditions where water discharge covaries with subglacial sediment transport capacity. In subglacial channels, the timing of peak water velocity and sediment transport capacity occurs before peak water discharge during a discharge event, due to evolving channel size. In subaerial channels, the timing of peak sediment transport capacity and water discharge coincide. Results here suggest that, even in a transport-limited subglacial system, a variable relationship between water and subglacial sediment discharge. Water discharge variations in both the Greenland Ice Sheet and Alpine cases are variable enough to cause this variable relationship. This variable relationship presents a challenge in linking hydro-climatic conditions or events to sediment export from glaciers. Water discharge and sediment transport capacity covariance could be possible when water discharge varies at a slower rate than subglacial channel size, such as Antarctica with minimal surface melt input. Results also suggest that water discharge records averaged over periods longer than $12$ hours from Alpine and Ice Sheet hydrographs show substantial impacts on subglacial sediment transport capacity characteristics."

- **407-10** It would not do any harm here to point to advances in subglacial sediment tracking, which speaks to this issue. I am well familiar with Jenkins' work – but there may well be others too.

*Excellent comment that we should have already accounted for. Jenkins et al. 2023 will be added and referenced.*

**References**

Perolo, P., Bakker, M., Gabbud, C., Moradi, G., Rennie, C., and Lane, S. (2018). Subglacial sediment production and snout marginal ice uplift during the late ablation season of a temperate valley glacier. *Earth Surface Processes and Landforms*, 0:1–68.

---

## Author Comment (AC2)

Dear Editor,

We thank Reviewer 2 for their comments. We believe that in addressing them, we have clarified the model presentation. We note, however, that the presentations of criticisms by the reviewer was at times difficult to respond to, given the format of the review. Please let us know if additional points should be addressed.

The reviewer's comments are in bold, our response is in italics and quotations from the new text are in normal font.

Best regards,

Ian Delaney on behalf of all authors

**General Comments**

- **The trouble is, that while the first term on the right hand side of (3) makes sense to me, the second does not, and the reason for this is that the closure term involves the effective pressure and thus the hydraulic head, but not its gradient, so I am a bit sceptical about this, because the structure of the Nye model seems to have been altered. You might say that the distinction disappears in the lumping, but it seems to me that if you have an upstream head h- and a downstream head h+, the first term on the right of (3) will involve the difference, but the second will involve the average, and there are two independent quantities involved. I suppose you can get around this by saying, oh, the outlet head or more accurately the effective pressure is zero, and perhaps that is what is done. But that is a bit disingenuous, because the outflow becomes open channel flow somewhere upstream of the actual mouth of the stream. Now I went and looked up this Werder 2010 paper, and you can see in its equation (4) the same distinction I am making here. I suppose my overall view is that (I think) the Clarke paper intended to use the flow elements of resistors, etc., as ingredients of a larger scale flow path, and this lumping of a whole channel is an extremely coarse thing to do, and looks a bit like avoiding confronting the spatially-dependent physics simply because it's too hard. So I think you can rescue this aspect of the model presentation, but a bit more exposition would help, in particular the figure 2or equivalent of the 2010 paper would help.**

  *We thank the reviewer for this comment, as it demands clarification in our model presentation. To reduce the number of variables in the original text, $h_p$ was omitted as its value is zero, as is the hydraulic head at the glacier terminus. The text has been updated to clarify this point. Additionally, we confirm the reviewer's comment, that effective pressure at the glacier terminus is zero. The text will read:* The evolution of subglacial channel size $S_g$ is given as

$$\frac{\partial S_g}{\partial t} = C_1 \frac{Q \Delta h}{l} - C_2 \left( h_o - \overline{h} \right)^n S_g, \tag{1}$$

  where $t$ is time, $C_1 = (1 - \rho_w c_p c_t) \frac{\rho_w g}{\rho_i L}$ and $C_2 = 2A(\frac{\rho_w g}{n})^n$ are constants (values in Table 1), $g$ is the acceleration due to gravity, $Q$ is water discharge, $\overline{h} = \frac{1}{2}(h + h_p)$ is the mean hydraulic head in the channel with $h_p$ being the proglacial hydraulic head equal to zero, $l$, $h_o = \frac{\rho_i}{\rho_w} h_{ice}$ is the mean ice overburden pressure expressed in meter water equivalent ($\rho_w$ is density of water; $\rho_i$ is density of ice), and $n$ is Glen's n (usually $n = 3$; Glen, 1955). The first term on the equation's right side represents the channel opening by frictional heating, while the following term represents channel closure from ice deformation.

*The reviewer also correctly points out that water de-pressurizes underneath the glacier, in agreement with the other reviewer, we have included a comment that we assume pressurized flow across the glacier bed, as is common in subglacial hydrology models, although this may not always be the case (Perolo et al., 2018).*

*Lastly, we have chosen a simple model, lumped element, or channel segment model here as we believe that it most clearly and concisely explains the dynamics that we wish to discuss. Our strong opinion is that the experiments that we have done with the model, across a range of hydrographs, demonstrate consistently the behavior of subglacial channels. Therefore, we do not believe that a more complex model is needed to show these processes. The limitation of this approach is acknowledged and discussed.*

- **And the model should come with some caveats, e. g., you could say that it is a simplistic and possibly unreliable model. To be fair, this is done in section 5.3, but that is too late.**

*The "Methods" section is largely meant to describe the model, as opposed to discussing its behavior or simplicity. In response to this comment, we will move the "Experiment limitations" to a section directly after the "Results" but before the "Discussion."*

- **One of the comments made is that because the sub-aerial model has an algebraic relation between width and discharge (equation 9), the relationship between velocity and discharge is also algebraic; but this is simply a choice of the model. In reality, the width of a channel will also evolve over (long) time through processes of bank erosion, and the use of equation 9 properly only involves a long term time average of discharge, so it is misleading to use it in a time-specific way. Actually, this is admitted at line 370, along with other limitations in the models.**

*We appreciate these comments. With regard to the first point about channel width evolution in the subaerial model, this matter is confronted in Sections 4.2 and 4.3. Here we show that across a range of exponents and relationships between channel cross-section and water discharge, subglacial systems are still more sensitive to water discharge variations than subaerial ones. This increased variability in subglacial sediment transport capacity occurs across a range of $\alpha$ values that describe the relationship between channel width and water discharge as shown in Figure 6 (will be 7). As a result, we believe that these results hold across a range of water discharge- channel width relationships, including ones where $\alpha$ evolves in time. The exception, shown in Section 4.3, is that when water discharge evolves more slowly than channel size. Here, the behavior between subaerial and subglacial is very similar, because channel areas in both channel types adjust to discharge variations.*

- **Two kinds of numerical experiments are described in section 3.3, and the results of these presented in section 4. Presumably the input to the model runs is the (time- varying) water discharge, but that seems not to be stated in 3.3, which I would expect, although it is stated in the caption to figure 2. The second half of the paper characterises the results of these experiments and there is an extended discussion. There is no doubt the authors have put a lot of effort into this, but my interest waned at this point.**

*To make it clear that water discharge is the input, we will modify the first sentence of Section 3.3 to explicitly state: "Proglacial discharge records from the Fieschergletscher (scenario ALPINE ) and the Leverett Glacier (scenario ICESHEET ) are used as inputs for the models above."*

- **In summary, this is one of those difficult papers to judge, because the effort involved is quite substantial and honestly applied, but the whole philosophy of the**

**approach is, in my view, misguided. This is a coxless boat crew who are rowing up a backwater without a rudder, and have lost the main direction of the stream.**

*With respect to the last sentence, it is true that the lead author feels this way occasionally. Regardless, we think several valuable and unique findings are within this manuscript.*

**Specific Comments**

- **33: I don't know what 'Following mass conservation' has to do with this.**

  *We will remove this phrase.*

- **99-100: upon that of Clarke.**

  *Done.*

- **102: presumably $h_{ice}$ is the thickness of the ice, but the syntax is not clear, it could be the depth of the channel.**

  *The text will read:* " a glacier with channel length $l$, with a flat bed and a mean ice thickness of $h_{ice}$."

- **eq. 3: it's probably in the Werder 2010 paper, but I'd like to see an extra comment about where this $\frac{\Delta h}{2}$ term comes from. Since the bracket represents $N = pi - pw$, $pw$ the pointwise form of this second term would be $\frac{pw}{\rho_w g}$, so it's not obvious to me where the $\Delta h$ comes from. Also in this light, it is worth elaborating the confusing term 'hydraulic head drop change' and defining what $\Delta h$ is in terms of $pw$. Actually, the more I think about this, the more suspicious I become. And in fact, at least when you're dealing with jökulhlaups, the discharge doesn't vary much along the length of the channel, and the consequence of that is that neither does the effective pressure. Of course, that may not apply here but I think the same principle applies. So maybe I am promoting this to a more substantive issue (as indeed now further discussed earlier).**

  *The comment about $\frac{\Delta h}{2}$ is addressed above. We understand the point about flow accumulation and increasing water discharge along the channel. As the author points out, the implications of this are presented in the "Experiment limitations." Additionally, we note that despite these effects, the results here show convincingly that the main topics in the paper still hold, i.e. sediment transport capacity is more variable in subglacial channels than subaerial ones and sediment transport capacity does not vary with water discharge if the subglacial channel size is out of equilibrium with water discharge.*

- **114:** I had no idea what the central angle referred to, but evidently it is the angle subtended at the centre of the circular arc of the channel upper boundary; and then $\beta = 2\alpha$, where $\alpha$ is the contact angle at the channel edge, which seems a more natural quantity to use. Also I checked the algebra of equation (5), and got this formula, but with the factor of two in the numerator, not the denominator, assuming hydraulic diameter is twice hydraulic radius, the latter of which is area/perimeter, so please check this. Incidentally, I did also then check equation (7) and I agree with that, so I do think there is an error in (5).

  *We thank the reviewer for the careful assessment of our work. We have double-checked and believe that there is no problem in the equation. Our proof is below:*

[Figure]

Guiding equations:

Hydraulic diameter:

$$D_h = \frac{4S_g}{P_w} \quad (I)$$

(2)

Wetter perimeter of the conduit (Hooke et al., 1990, pg. 69):

$$P_w = R\beta + 2R\sin\frac{\beta}{2} = R(\beta + 2\sin\frac{\beta}{2}) \quad (II)$$

(3)

Cross sectional area of the conduit (Hooke et al., 1990, pg. 69):

$$S_g = \frac{R^2}{2}(\beta - \sin\beta) \quad (III)$$

(4)

$D_h$ is hydraulic diameter, $P_w$ is wetter perimeter, $R$ is radius, $\beta$ is Hooke angle (Hooke et al., 1990), and $S_g$ is the subglacial channel cross section.

Proof:

$$\frac{D_h}{4S_g} = \frac{1}{R\beta + 2R\sin\frac{\beta}{2}} \quad I\text{into } II$$

(5)

$$R = \frac{4S_g}{D_h(\beta + 2\sin\frac{\beta}{2})}, \quad \text{into } III$$

(6)

$$S_g = \frac{16S_g^2}{D_h^2(\beta + 2\sin\frac{\beta}{2})^2} \quad \frac{1}{2}(\beta - \sin\beta)$$

(7)

$$1 = \frac{8S_g}{D_h^2(\beta + 2\sin\frac{\beta}{2})^2} \quad (\beta - \sin\beta)$$

(8)

$$\frac{D_h^2}{8} \quad \frac{(\beta + 2\sin\frac{\beta}{2})^2}{\beta - \sin\beta} = S_g$$

(9)

$$\frac{D_h^2(2(\frac{\beta}{2} + \sin\frac{\beta}{2}))^2}{8(\beta - \sin\beta)} = S_g$$

(10)

$$\frac{D_h^2(\frac{\beta}{2} + \sin\frac{\beta}{2})^2}{2(\beta - \sin\beta)} = S_g$$

(11)

$$S_g = \frac{D_h^2}{2}\frac{(\frac{\beta}{2} + \sin\frac{\beta}{2})^2}{(\beta - \sin\beta)}$$

(12)

This is the same as Equation 5 of our manuscript.

**References**

Glen, J. (1955). The creep of polycrystalline ice. *Proceedings of the Royal Society of London. Series A. Mathematical and Physical Sciences*, 228(1175):519–538.

Hooke, R. L., Laumann, T., and Kohler, J. (1990). Subglacial water pressures and the shape of subglacial conduits. *Journal of Glaciology*, 36(122):67–71.

Perolo, P., Bakker, M., Gabbud, C., Moradi, G., Rennie, C., and Lane, S. (2018). Subglacial sediment production and snout marginal ice uplift during the late ablation season of a temperate valley glacier. *Earth Surface Processes and Landforms*, 0:1–68.

---

## Referee Report (RR1)

Sediment transport capacity response to variations in water discharge in pressurized subglacial channels (Revision 1)

Author(s): Delaney and others

**General comments**

The revised manuscript is improved and, while the multivariate nature of the analysis prevents the main findings from being expressed as clearly and succinctly as I would consider ideal, I have no substantial issues. As well as a few very minor comments below, I have two more general suggestions. First, many of the Figures still need attention in terms of consistent font and case, use of colours (especially font on a coloured background), and accurate and consistent labelling and captioning. Second, the point made in lines 17 – 19 of the Abstract is still a bit bland. If possible, I would rather make an explicit point that (since sediment transport capacity and competence are more variable, and therefore include higher peaks, for a given discharge in subglacial settings) former subglacial discharge reconstructed from the texture of remnant 'flood' deposits may well be far too high. Perhaps this could even be quantified and presented as a headline figure (peak tau is 1M higher in ICESHEET subglacial than subaerial in Fig. 2g, no?). Or just note this as 'For example…'.

**Specific comments**

| Line/Location | Comment/Suggestion |
| --- | --- |
| | |
| Fig 1 | Fill and font colours don't work very well; axis legend capitalization is inconsistent with elsewhere |
| Fig. 1 caption | 'Sketch of…' |
| 57 | 'Yet, continually evolving channel size can…' Incidentally, I'd check all uses of 'evolve/evolving' through the ms and look to replace some with 'change/changing' (e.g., I would use changing here [and e.g., line 154]). |
| 95 | I would replace 'upon' with 'on'. Check elsewhere too. |
| 109 | Italicise '$n$'? |
| 136 | Delete 'the' (twice) as both are names/proper nouns. |
| 139 | Delete 'thickness' |
| Fig. 2 caption | Last line doesn't make sense (but the grey band along e – h does need explanation). |
| 205 | '…occurs because diurnal…'. In fact, I think I'd combine to ''…increase in correlation (Figure 4), reflecting the reduced diurnal variability" |
| Fig. 3 | Axis legend capitalization |
| Fig. 3 caption | 'Relationships between…' Also, insert a space between '15' and 'min' |
| Y axis | I'd note that the axis is logarithmic (here and elsewhere). The issue arises because there are no intermediate ticks here and only two major markers so the reader does not know whether the axis scale is normal or log. Maybe just insert minor tick marks (as is done, albeit variably, elsewhere). |
| 208-9 | This sentence is really difficult to follow… |
| 212 | '…higher … than…' (not '…higher … compared to…' (elsewhere too) |
| Figure 5 | There are 9 dots on each plot but only 8 values noted in the caption (2 d missing from the latter?) Comma missing after '6 hr' in the caption too. |

| | |
|---|---|
| 226 | '…shear stress, which can…' |
| 227 | '…stress, but…' |
| Figure 6 caption | I'm pretty sure 'left' and 'right' are the wrong way around here… Also, '…show the range…' |
| 321 | '…transport capacity and water discharge…' |
| 330 | 'The last of these can remain relatively stable over years…' |
| 333 | '…likely largely represents…' (?) |
| 362-3 | '…melt is too short for the channel to reach steady state' needs rewriting. Maybe '…when high-frequency variability in melt rate is too great for…' |
| 366 | 'Thus, fluctuations in discharge…' |
| 369 | '…these variations in sediment transport capacity are far larger than those…' |
| 374 | '…or the threshold at which…' |
| 287 | Delete 'exists' |
| 389 | Replace 'as' with 'because' |
| 400 | Replace 'its' with 'channel' |
| 406 and 417 | I'd remove reference to objectives and begin with 'We established the conditions under which… ' etc |
| 408 | '…and an ice sheet…' |
| 410 | '…a variable relationship exists between water discharge and …' |
| 411 | I'd delete 'potentially' |
| 412 | '…even when those records are smoothed…' |
| 418 | It is not clear what variables are higher than what else. |
| 420 | '…variations, as in the…' |

---

## Author Response (AR2)

Dear Editor,

We thank Reviewer 1 for their detailed comments. This last round of edits has helped with many details in the manuscript.

The reviewer's comments are in bold, our response is in normal font and quotations from the new text are in italic font.

Best regards,

Ian Delaney on behalf of all authors

**General Comments**
**The revised manuscript is improved and, while the multivariate nature of the analysis prevents the main findings from being expressed as clearly and succinctly as I would consider ideal, I have no substantial issues. As well as a few very minor comments below, I have two more general suggestions.**

- **First, many of the Figures still need attention in terms of consistent font and case, use of colours (especially font on a coloured background), and accurate and consistent labelling and captioning.**

  This comment point to the careful and detailed examination by the reviewer and we are very grateful for this. In addition to making the changes detailed below in the specific comments, we have carefully examined each of the figures to ensure they are consistent and match the text and caption.

- **Second, the point made in lines 17 – 19 of the Abstract is still a bit bland. If possible, I would rather make an explicit point that (since sediment transport capacity and competence are more variable, and therefore include higher peaks, for a given discharge in subglacial settings) former subglacial discharge reconstructed from the texture of remnant 'flood' deposits may well be far too high. Perhaps this could even be quantified and presented as a headline figure (peak tau is 1M higher in ICESHEET subglacial than subaerial in Fig. 2g, no?). Or just note this as 'For example...'**

  We thank the reviewer for this comment. While we understand and are very grateful for the reviewer's recommendations, we have only slightly modified the text by removing the referenced lines and changing the following sentence. *The implications of these findings help to evaluate sediment discharge from glaciers with different hydro-climatic forcings and establish sources of variability in the sediment export-water discharge relationships.* The implications of this study could be used to interpret sediment transport during flooding events, as proposed by the reviewer. However, we are hesitant to mention this directly in the abstract as it remains uncertain if such deposits would be indicative of subaerial sediment transport conditions once the water leaves the glacier or subglacial ones.

  In response to the part of the comment referring to larger difference in subglacial compared to subaerial sediment transport capacity, we have added the following sentence to the second to last paragraph of Section 6.2. *In addition to greater variability, larger sediment transport capacity in subglacial systems compared to subaerial ones (Figure 2) could cause sediment deposition at the glacier margin in the transition from pressurized to open channel flow (Perolo et al., 2018; Mancini et al., 2023; Delaney et al., 2024).*

**Specific Comments**

- **Fig 1 Fill and font colours don't work very well; axis legend capitalization is inconsistent with elsewhere**

  *variables in black remain constant with short variations in water discharge, where as variables in color change.*

- **Fig. 1 caption 'Sketch of...'**

  Done.

- **57 'Yet, continually evolving channel size can...' Incidentally, I'd check all uses of 'evolve/evolving' through the ms and look to replace some with 'change/changing' (e.g., I would use changing here [and e.g., line 154]).**

  The matter is fixed. We have kept all references to "evolving channel size" in the text given the rational in the previous version, whereby channel size depend on antecedent conditions (i.e. Equation 3). However, other uses of "evolving" have been changed.

- **95 I would replace 'upon' with 'on'. Check elsewhere too.**

  We believe that the use of "upon" here is appropriate.

- **109 Italicise 'n'?**

  Done.

- **136 Delete 'the' (twice) as both are names/proper nouns.**

  Done.

- **139 Delete 'thickness'**

  Done.

- **Fig. 2 caption Last line doesn't make sense (but the grey band along e – h does need explanation).**

  Done. We have also removed the gray bands associated with the heatwave.

- **205 '...occurs because diurnal...'. In fact, I think I'd combine to '' ...increase in correlation (Figure 4), reflecting the reduced diurnal variability"**

  The text now reads:

  *Smoothing water discharge over periods longer than one day causes a substantial increase in rank correlation (Figure 4), removing diurnal variations (Figures S1-S14). Even with the variations removed, correlations in sediment transport characteristics in ICESHEET remain higher than ALPINE, which has more water discharge variations even when smoothed compared to ICESHEET (Figure S1-S14).*

- **Fig. 3 Axis legend capitalization**

  Done.

- **Fig. 3 caption 'Relationships between...' Also, insert a space between '15' and 'min' Y axis I'd note that the axis is logarithmic (here and elsewhere). The issue arises because there are no intermediate ticks here and only two major markers so the reader does not know whether the axis scale is normal or log. Maybe just insert minor tick marks (as is done, albeit variably, elsewhere).**

  We have addressed the first two comments. However, the Y-axis is normal scale, not log, so we do not see the issue.

- **208-9 This sentence is really difficult to follow...**

  Text now reads *The second numerical experiment aims to compare the variability between the subglacial and subaerial sediment transport capacities. To do this, model outputs from both systems are forced with a range of channel shapes, friction factors, and water discharge variability.*

- **212 ' ...higher ... than... ' (not '...higher ... compared to...' (elsewhere too)**

  Done.

- **Figure 5 There are 9 dots on each plot but only 8 values noted in the caption (2 d missing from the latter?) Comma missing after '6 hr' in the caption too.**

  Thank you for the careful read. the $2\,\mathrm{d}$ period was missing. Comma added.

- **226 '...shear stress, which can...'**

  Done.

- **227 '...stress, but...'**

  Done.

- **Figure 6 caption I'm pretty sure 'left' and 'right' are the wrong way around here... Also, '...show the range...'**

  Done.

- **321 '...transport capacity and water discharge...'**

  Done.

- **330 'The last of these can remain relatively stable over years...'**

  Done.

- **333 '...likely largely represents...' (?)**

  Done.

- **362-3 ' ...melt is too short for the channel to reach steady state' needs rewriting. Maybe '...when high-frequency variability in melt rate is too great for...'**

  Changed to: *especially during severe rain or melt where melt water input timescales are too*

- **366 'Thus, fluctuations in discharge...'**

  Done.

- **369 '...these variations in sediment transport capacity are far larger than those...'**

  Done.

- **374 '...or the threshold at which...'**

  Done.

- **287 Delete 'exists'**

  Done.

- **389 Replace 'as' with 'because'**

  Done.

- **400** Replace 'its' with 'channel'

  Done.

- **406 and 417**  I'd remove reference to objectives and begin with 'We established the conditions under which… ' etc

  Given the explicit statement of goals and objectives in the introduction, we would like to keep this reference to the objectives of the study.

- **408** '…and an ice sheet…'

  Done.

- **410** ' …a variable relationship exists between water discharge and …'

  Done.

- **411**  I'd delete 'potentially'

  Done.

- **412** '…even when those records are smoothed…'

  Done.

- **418**  It is not clear what variables are higher than what else.

  Done.

- **420** '…variations, as in the…'

  Done.

**References**

Delaney, I., Werder, M., Felix, D., Albayrak, I., Boes, R., and Farinotti, D. (2024). Controls on Sediment Transport From a Glacierized Catchment in the Swiss Alps Established Through Inverse Modeling of Geomorphic Processes. *Water Resources Research*, 60(4):e2023WR035589.

Mancini, D., Dietze, M., Müller, T., Jenkin, M., Miesen, F., Roncoroni, M., Nicholas, A., and Lane, S. (2023). Filtering of the Signal of Sediment Export From a Glacier by Its Proglacial Forefield. *Geophysical Research Letters*, 50(21):e2023GL106082. e2023GL106082 2023GL106082.

Perolo, P., Bakker, M., Gabbud, C., Moradi, G., Rennie, C., and Lane, S. (2018). Subglacial sediment production and snout marginal ice uplift during the late ablation season of a temperate valley glacier. *Earth Surface Processes and Landforms*, 0:1–68.